

# Best practice regarding the three P's: profiling, portability and provenance when running HPC geoscientific applications

Wendy Sharples[1,2,3], Ilya Zhukov[1], Markus Geimer[1], Klaus Goergen[2,4], Stefan Kollet[2,4], Sebastian Luehrs[1], Thomas Breuer[1], Bibi Naz[2,4], Ketan Kulkarni[1,4], and Slavko Brdar[1,4]

[1]Jülich Supercomputing Centre, Research Centre Jülich, Jülich, Germany
[2]Institute of Bio- and Geosciences, Agrosphere (IBG-3), Research Centre Jülich, Jülich, Germany
[3]Meteorological Institute, University of Bonn, Bonn, Germany
[4]Centre for High-Performance Scientific Computing in Terrestrial Systems, Geoverbund ABC/J, Jülich, Germany

*Correspondence to:* Wendy Sharples (w.sharples@fz-juelich.de)

**Abstract.** Geoscientific modeling is constantly evolving, with next generation geoscientific models and applications placing high demands on high performance computing (HPC) resources. These demands are being met by new developments in HPC architectures, software libraries, and infrastructures. New HPC developments require new programming paradigms leading to substantial investment in model porting, tuning, and refactoring of complicated legacy code in order to use these resources effectively. In addition to the challenge of new massively parallel HPC systems, reproducibility of simulation and analysis results is of great concern, as the next generation geoscientific models are based on complex model implementations and profiling, modeling and data processing workflows.

Thus, in order to reduce both the duration and the cost of code migration, aid in the development of new models or model components, while ensuring reproducibility and sustainability over the complete data life cycle, a streamlined approach to profiling, porting, and provenance tracking is necessary. We propose a run control framework (RCF) integrated with a workflow engine which encompasses all stages of the modeling chain: 1. preprocess input, 2. compilation of code (including code instrumentation with performance analysis tools), 3. simulation run, 4. postprocess and analysis, to address these issues. Within this RCF, the workflow engine is used to create and manage benchmark or simulation parameter combinations and performs the documentation and data organization for reproducibility. This approach automates the process of porting and tuning, profiling, testing, and running a geoscientific model. We show that in using our run control framework, testing, benchmarking, profiling, and running models is less time consuming and more robust, resulting in more efficient use of HPC resources, more strategic code development, and enhanced data integrity and reproducibility.

## 1 Introduction

Geoscientific modeling is constantly evolving, leading to higher demands on HPC resources. We distinguish four main developments which increase HPC demands. (i) Higher spatial resolution, where the added value inherent to simulations at high spatial resolutions has been shown for example in many studies of regional convection permitting climate simulations (e.g., Kendon et al., 2017; Prein et al., 2015; Eyring et al., 2015; Heinzeller et al., 2016) and continental hyper-resolution hydrological





modeling approaches as demonstrated by (Kollet and Maxwell, 2008; Maxwell et al., 2015; Keune et al.); (ii) increased model domain size, where models are now being run at larger scales, for example, global convection permitting models (Schwitalla et al., 2016), high resolution continental RCMs (Leutwyler et al., 2016) and global hydrology and land surface models which are needed for water resources modeling (Bierkens et al., 2015); (iii) increased model complexity in which the desire to ex-

plore the feedbacks between the surface, subsurface, oceans, and atmosphere have led to fully coupled multi-physics global or regional earth system models (ESM) (Hazeleger et al., 2010; Hurrell et al., 2013; Ruti et al., 2016; Shrestha et al., 2014) posing load balancing issues (Gasper et al., 2014), and (iv) increasing number of ensemble members in modeling and data assimilation experiments (e.g., Gutowski Jr. et al., 2016; Han et al., 2016; Kurtz et al., 2016). These developments, combined with long climate scenario simulation time spans pose specific challenges in terms of computational resources, data volume,

data velocity, data handling, and analysis.

The evolution of geoscientific models is closely linked to and enabled by technical advancements in supercomputing, for example, advances in computer chip development, low cost growth in parallelism, low latency high bandwidth interconnects, or parallel file systems (e.g., Smari et al., 2016; Navarra et al., 2010). Current HPC systems are massively parallel distributed memory supercomputers. Shared memory compute nodes with two or more multi-core CPUs are linked with low latency inter-

connects (Beanato et al., 2013). Hybrid parallelization, combining multithreading (e.g., via OpenMP or POSIX threads) on a shared memory node and inter-node distributed memory parallelism (via Message Passing Interface, MPI) reduces communication overhead, improves partitioning, load balancing, and communication overlapping as well as memory footprint (Jin et al., 2011). The inter-process communication can be further optimized by explicitly mapping threads onto specific nodes and CPU cores taking into account the interconnect topology and the model's internal communication pattern (Balaji et al., 2011).

HPC hardware, software, and tools are developing at a rapid pace. For example, heterogeneous HPC architectures that combine multi-core CPUs with accelerators on the same compute node (Brodtkorb et al., 2010) are considered a suitable architecture for future exascale systems, that is, supercomputers with more than $10^{18}$ floating-point operations per second, because of their energy efficiency (i.e., Flops/Watt), low latency data management, and peak performance per accelerator of currently more than 1 TFLOP/s (Davis et al., 2012). Accelerator design is currently either based on graphic processing units

(GPUs) or on Many Integrated Core chips (MICs). These coprocessors have tens of cores and can host hundreds of threads per chip, include their own memory with very high bandwidth (Liu et al., 2012), albeit using different memory architectures (e.g., cache coherence) or parallel programming models (e.g., CUDA, OpenCL, OpenACC). While these HPC developments are instrumental towards next-generation exascale HPC systems, during the next decade (Attig et al., 2011; Keyes, 2011; Davis et al., 2012), MPI-parallel simulation codes on multi-core shared or distributed memory architectures need a substantial

amount of porting, profiling, tuning, and refactoring (Hwu, 2014) to efficiently use such kind of hardware, in particular because a very high level of vectorization is needed to take advantage of the ever increasing SIMD units. Moreover, offloading compute intensive code sections to accelerators can also become a performance bottleneck due to excessive data transfers between host and accelerator. Thus, special care needs to be taken with respect to data layout, placement, and reuse. Concurrent to improvements in the simulation codes themselves, development of established parallel solver libraries such as SUNDIALS

and PETSc has been underway to enable the use of heterogeneous architectures (e.g., Minden et al., 2013; Breß and Saake,





2013; Kraus et al., 2013). In addition, compilers are constantly improved to provide some level of vectorization when using appropriate optimization flags (Petersen et al.), though additional effort may be required to structure the source code such that compilers are enabled to recognize vectorization opportunities.

In the geosciences, software applications are often complicated by the new developments outlined above. In many cases, 5 complicated legacy codes need substantial investments in model porting, tuning, and refactoring in order to efficiently use these upcoming and already existing HPC architectures, software libraries, and infrastructures and achieve a high level of performance. Invested effort has already paid off for many codes (Meadows, 2012; Hammond et al., 2014; Leutwyler et al., 2016; Heinzeller et al., 2016), however, at a significant cost in resources. In order to reduce both the duration and the cost of code migration, and also aid in the development of new models or model components, a systematic, rigorous approach is needed 10 to fully analyze and understand the runtime behavior and I/O characteristics in detail, and identify performance bottlenecks. In this context, the use of performance analysis tools is crucial. However, depending on the current focus of the analysis, different tools and techniques may have to be used—sometimes even in combination. For example, while various tools provide generic information about the runtime behavior of an application, specialized tools exists that focus on a particular aspect such as vectorization, threading, communication and synchronization, or I/O. Likewise, while profiling—i.e., the process-local 15 generation of aggregated performance metrics during the execution—can provide a summarized overview of the performance for the entire application run, it is not able to capture the dynamic runtime behavior. Thus, it can be complemented by using event tracing, which collects performance-related events in chronological order and therefore allows to reconstruct the dynamic application behavior in detail. However, care has to be taken when using event tracing, as it is more expensive than profiling as the amount of data in the trace increases with the runtime of the application (e.g., Geimer et al., 2010; Carns et al., 2011). 20 Thus, it is usually only applied to selected parts of the execution, for example, a few time steps or iterations of a solver, that have been identified using more lightweight techniques such as profiling.

In addition to making efficient use of massively parallel HPC systems, reproducibility of simulation results, based on complex model implementations, profiling, modeling, and data processing workflows, must be a fundamental principle in computational research (Hutton et al., 2016). Recently, Stodden et al. (2016) presented "Reproducibility Enhancement Principles" 25 (REP) to help ensure that the computational steps in data processing and generation are similarly important as access to the data themselves. Hence sharing not only data but also details on software, workflows, and the computational environment via open repositories is likewise important. Similarly, Hutton et al. (2016) recommend for computational hydrology that workflows, which combine data and reusable code, are needed in order to ensure provenance of scientific results. Given that in the weather and climate sciences data and primary code availability is often ensured, ancillary code availability is addressed in 30 Irving (2016) as one of the root causes for irreproducibility. With this in mind, we consider the aspect of documenting the porting and performance optimization steps as well as provenance tracking during production simulations as highly relevant to ensure reproducibility. Workflow engines such as ecFlow (Lin et al., 2015) or Cylc (Oliver et al., 2017) can connect all relevant steps of a modeling chain, submit jobs with dependencies, and help with necessary parameter sweeps for application software porting and tuning alike. At the same time they allow for extensive, systematic logging of the processing steps themselves as 35 well as the log outputs from the individual applications.





In this article, we present a streamlined run control framework (RCF) to porting, profiling, and documenting legacy code using the script-based benchmarking framework JUBE (Lührs et al., 2016) as a workflow engine. We developed profiling, run control, and testing frameworks which are dynamically built with user input and run using JUBE. While the use case for this portable run control, profiling, and testing workflow engine system discussed in this paper is the software application

ParFlow, an integrated parallel watershed model, the RCF is generic and can be applied to any other simulation software. In the remainder of this paper we outline this approach and highlight the subsequent developments scheduled for implementation born out of the extensive profiling of ParFlow. Additionally, we highlight other uses for using a workflow engine which have enabled us to streamline the run control process for model runs.

## 2 RCF approach to porting, profiling, and provenance tracking

In this section, the run control framework which could be described as a run harness, along with JUBE is introduced, where a harness in this case is used to describe the framework of scripts and other supporting tools that are required to execute a workflow. This is followed by the description of the standard profiling toolset which is currently built into our RCF to aid in the ParFlow hydrological model development and for production simulation run control. The numerical experiment test case, which is used as a baseline for the profiling and model improvement, is presented alongside the hardware characteristics of the

supercomputers used in this study. The RCF for the case study is given in the Appendix B.

### 2.1 JUBE as a workflow engine

Benchmarking scientific code could assess impacts of changes of the underlying HPC software stack (e.g., compiler or library upgrades) and hardware (e.g., interconnect upgrades), aid in testing as part of software engineering and code refactoring, and finding optimum numerical model configurations. Benchmarking a numerical model system usually involves several runs with

different configurations (compiler, domain, physics parameterizations, solver settings, load balancing), including compilation, instrumentation (i.e., the injection of special monitoring "hooks" into the program to enable profiling and/or event tracing), various simulations, profiling, result verification, and analysis. However, with increasing model and HPC environment complexity, the parameter space for benchmarking can be large. To avoid errors in managing benchmark parameter combinations, to reduce the overall temporal effort, and to ensure reproducibility and comparability, benchmarking must be automated. This task can

be accomplished using a workflow engine, which is an application for workflow automation, like the JUBE benchmarking environment (Lührs et al., 2016).

JUBE is a script-based framework designed to efficiently and systematically define, setup, run, and analyze benchmarks and production simulations. The current JUBE v2.1.4 is a Python-based implementation released under GNU GPLv3 actively developed at the Jülich Supercomputing Centre (JSC, https://www.fz-juelich.de/ias/jsc/EN). JUBE allows to easily define benchmark

sets in which the workflow and parameter sweeps are specified by an XML configuration file. JUBE controls the automatic execution of the designed workflow and takes care of the underlying file structure to allow an individual execution per run. Automatic bookkeeping separates the different runs and parameter combinations and allows reproducible executions. To gen-





erate an overview of the overall workflow execution, the user can configure JUBE how to analyze the different output files to extract information such as the overall runtime or other application-specific data. This allows the system to create a combined overview of the underlying parametrization and the application outputs.

Leveraging the generic JUBE framework, we developed a run control framework (RCF), suitable for a typical geoscience
model, from a series of XML files integrated with Python scripts to be executed with JUBE (see Figure 1 and Appendix B). These jobs are usually run with the following modeling chain: 1. preprocess input, 2. compilation of code (includes code instrumentation with profiling tools) 3. simulation run 4. postprocess and analysis. The current run control framework is under version control and can be cloned from GitLab (Hethey, 2013). Details about the individual steps in this modeling chain are outlined in the following subsections.

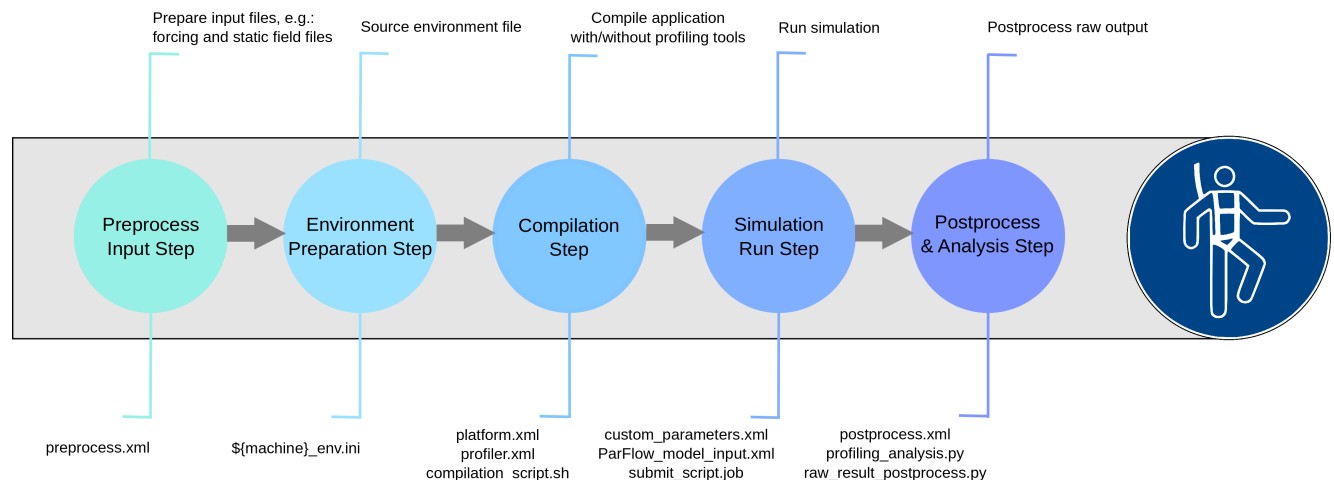

**Figure 1.** Schematic overview of the modeling chain as supported by our JUBE-based run harness. Each step is annotated with a brief description (top) as well as the respective RCF infrastructure (XML files and scripts, bottom).

**2.1.1   Code portability**

Within the RCF, we have separated all the information pertinent to the existing compilers, required environment modules, and workload manager job submission specifics for a given system into a single XML file (see Figure 1, `platform.xml` and Appendix B2.2). This `platform.xml` file can easily be extended to include any new system. When compilers, environment modules, and workload managers are updated or new features or functionality are added, the `platform.xml` can be easily
altered to include these updates. For example, as new C and Fortran compiler versions are released with improved code generation and potentially new optimization flags, it is useful to reassess which compilation flags give the best run time without compromising the accuracy of the result. In the case of our RCF, this is as simple as altering the compiler flags parameter in `platform.xml` with comma separated values for each different compiler flag, which produces a benchmark suite (see





Appendix B2.2). In order to ensure accuracy is not compromised we built in a result comparison test in the postprocessing and analysis step, in order to compare the result with previously generated output.

### 2.1.2 Code profiling

In order to analyze ParFlow's runtime behavior, determine optimal runtime settings, as well as identify performance bottlenecks during model development, we use several complementary performance analysis tools. Set up, compilation wrappers, and analysis profiling steps were built into our RCF (Figure 1: `profiler.xml`, Appendix A) with support for the following tools: Score-P v3.1 (Knüpfer et al., 2012) and Scalasca v2.3.1 (Geimer et al., 2010; Zhukov et al., 2015), where results collected with Score-P and Scalasca can be examined using the interactive analysis report explorer Cube v4.3.5 (Saviankou et al., 2015), Allinea Performance Reports v7.0.4 (January et al., 2015), Extrae v3.4.3 (Alonso et al., 2012), Paraver v4.6.3 (Labarta et al., 2006), Intel Vectorization Advisor 2015 (Rane et al., 2015), and Darshan v3.0.0 (Carns et al., 2011) (see Table A1 in Appendix A for a more detailed description of each performance analysis tool listed above). The modeling chain for the profiling workflow is as follows:

1. Prepare the input data;

2. Load environment modules and set up performance analysis tool specific parameters;

3. Compile or link ParFlow using scripts and wrappers, depending on what is required by the profiling tool; e.g., Score-P requires compilation and linking using wrappers, whereas Darshan requires only linking after compilation;

4. ParFlow execution with the necessary tool flags (e.g., Scalasca has various runtime measurement, collection, and analysis flags which can be turned on or off);

5. Parse and analyze the results interactively (e.g., using interactive visual explorers like Paraver, Cube, etc.) or generate a performance metric report via a post processing step.

Note that code instrumentation with performance analysis tools—that is, the insertion of tool-specific measurement calls into the application code which are executed at relevant points (events) during runtime—can introduce significant overhead, which can be assessed by comparing to an uninstrumented reference run. If the runtime of the instrumented version of the code under inspection is much longer than the reference run (more than 10-15%), it is recommended to reduce instrumentation overhead as the measurement may no longer reflect the actual runtime behavior of the application. Typical measures to reduce the runtime overhead include turning off automatic compiler instrumentation, filtering out short but frequently called functions, and applying manual instrumentation using specific APIs provided by the tools.

A typical workflow when performing an initial "health examination" on a scientific code can be described as follows:

1. "Which function(s) or code region(s) in my program consume(s) the most wallclock time?" This question can usually be answered using a flat profile, which breaks down the application code into separate functions or manually annotated



source code regions (e.g., initialization vs. solver phase) and aggregates the wallclock time spent in each function/region. This ascertains the area(s) of interest in order to streamline performance analysis efforts.

*Typical diagnosis tools: Allinea Performance Reports, Score-P + Cube, Extrae + Paraver*

2. "Does my application scale as expected?" Typically all scientific applications aim to perform well at scale. To address this question, profiles need to be collected with varying numbers of processes and the scalability of functions/code regions within the areas of interest can be examined.

   There are two types of scaling: strong and weak scaling. In case of strong scaling, the overall problem size (workload) stays fixed but the number of processes increases. Here, the runtime is expected to decrease with increasing number of processes. By contrast, in case of weak scaling, the workload assigned to each process remains constant with an increase of processors and, thus, the runtime is ideally expected to be constant as well.

   *Typical diagnosis tools: Allinea Performance Reports, Score-P + Cube, Extrae + Paraver*

3. "Does my program suffer from load imbalance?" If this is the case, some processes will be performing significantly more or less work than the others. Load balance is an indication of how well the load is distributed across processors. If a code is not well balanced, HPC resources will be used inefficiently as imbalances usually materialize as wait states in communication/synchronization operations between processes/threads. Thus, this may be an area to concentrate code refactoring efforts.

   Note that load imbalance can either be static or dynamic. While the former can usually be easily identified in profiles, pinpointing the latter may require more heavyweight measurement and analysis techniques such as event tracing, as imbalances may cancel out each other in aggregated profile data.

   *Typical diagnosis tools: Score-P + Cube, Score-P + Scalasca + Cube, Extrae + Paraver*

4. "Is there a disproportionate time spent in communication or synchronization?" Communication and synchronization overheads can be caused by network latencies (e.g., due to an inefficient process placement onto the compute resources), or wait states and other inefficiency patterns (e.g., caused by load or communication imbalances). If these overheads rise significantly with increase in resources, this can be a further barrier to scalability.

   *Typical diagnosis tools: Allinea Performance Reports, Score-P + Scalasca + Cube, Extrae + Paraver*

5. "Is my application limited by resource bounds?" There are several bounds one can reach, such as

   (a) CPU bound, i.e., the rate at which processes operate is limited by the speed of the CPU. For example, a tight loop that can be vectorized and operates only on a few values that can be held in CPU registers is likely to be CPU bound.

   (b) Cache bound, i.e., the simulation is limited by the amount and the speed of the cache available. For example, a kernel operating on more data than can be held in registers but which fits into cache is likely to be cache bound.





(c) Memory bound, i.e., the simulation is limited by the amount of memory available and/or the memory access bandwidth. For example, a kernel operating on more data than fits into cache is likely to be memory bound.

(d) I/O bound, i.e., the simulation is limited by the speed of the I/O subsystem. For example, counting the number of lines in a file is likely to be I/O bound.

*Typical diagnosis tools: Score-P + PAPI + Cube, Extrae + Paraver, Intel Vectorization Advisor, Darshan*

6. There are additional questions one can add to the survey, for example: "How many pipeline stalls, cache misses, and mis-predicted branches are occurring?", ""What is the Instructions Per Cycle (IPC) ratio?", etc.

    *Typical diagnosis tools: Score-P + PAPI + Cube, Extrae + Paraver, Intel Vectorization Advisor, Darshan, etc.*

The typical diagnosis tools mentioned in the above section are implemented in the RCF for various HPC platforms (see
Appendix A and B for more details).

The results shown in this study are obtained using the diagnosis tools available on the JUQUEEN platform, namely 1. Score-P profile measurements, including hardware performance counters collected via PAPI (Moore et al.), 2. Score-P trace measurements followed by a subsequent automatic Scalasca trace analysis, 3. Manual analysis of measurements from the abovementioned steps with an interactive visual browser, i.e., Cube. Additionally we used Darshan for I/O profiling. As Alinea Perfor-
mance Reports and Intel Vectorization Advisor are not available for the Blue Gene/Q platform, it was not possible to use them in the analysis. However, if they are available it is recommended to use them as they can provide additional valuable insight into the potential bottlenecks of the application.

In order to track the health of ParFlow with increasing simulation time, we developed an automated performance metric extraction workflow to obtain key performance indicators such as MPI wait time, memory footprint, cache intensity, etc. in
order to quickly assess whether new developments or additions to the code improve or degrade the overall performance. An example of such a workflow output is given in Figure 2.

### 2.1.3 Provenance tracking

JUBE has many provenance tracking features and tools. JUBE automatically stores the benchmark suite data for each workflow execution, which can be parsed by JUBE's analysis tools. Workflow metadata is automatically parsed by JUBE and then
compiled into a report detailing the run and which settings were used for each suite. Subsequent analysis procedures can be predefined, added, or altered by the user after the experiment to automate data processing. These features and tools are designed to facilitate documentation and archiving. Additionally, JUBE's workflow execution directory structure allows for run time provenance tracking. JUBE's workflow management system automatically creates a suite of the parameter sets and steps for each workflow. JUBE then creates a unique execution unit or work package for a specific step and parameter combination.
Each workflow execution has its own directory named by a unique numeric identifier which gets incremented for subsequent runs. Inside this directory, JUBE handles the workflow execution's metadata and creates a directory for each separate work



```
result:
                time[s] |          4
             time_io[s] |        0.1
            time_mpi[s] |        0.7
             mem_vs_cmp |        1.0
       load_imbalance[%] |        9.0
           io_volume[MB] |      183.1
               io_calls |          0
      io_throughput[MB/s] |     1624.4
       avg_io_ac_size[kB] |        0.0
          num_p2p_calls |      14128
        p2p_comm_time[s] |        0.4
     p2p_message_size[kB] |        8.1
          num_coll_calls |       1008
        coll_comm_time[s] |        0.2
    coll_message_size[kB] |        0.4
            delay_mpi[s] |        0.4
        delay_mpi_ratio[%] |       59.3
             time_omp[s] |
              omp_ratio[%] |        nan
            delay_omp[s] |
        delay_omp_ratio[%] |        nan
        memory_footprint |    92228kB
      cache_usage_intensity |       0.65
                     IPC |       1.53
           time_no_vec[s] |          5
                 vec_eff |       1.25
           time_no_fma[s] |          4
                 fma_eff |       1.00
```

**Figure 2.** Example output from a performance metric extraction workflow. Where time[s]: application total wallclock time, will be considered as a reference run, time_io[s]: average time spent in input/output operations for each rank, time_mpi[s]: average time spent in MPI for each rank, mem_vs_comp: memory vs. compute bound (close to 1.0 means strongly compute bound, close to 2.0 means strongly memory bound), load_imbalance: ratio of the load imbalance overhead towards the critical path duration, io_volume[MB]: total amount of data in I/O, io_calls[nb]: total number of I/O calls, io_throughput[MB/s]: speed of I/O, avg_io_ac_size[kB]: average amount of data per I/O call, num_p2p_calls[nb]: average number of point-to-point MPI operations per MPI rank, p2p_comm_time[s]: average time spent in point-to-point MPI operations per MPI rank, p2p_message_size[kB]: average size of point-to-point MPI messages per MPI rank, num_coll_calls[nb]: average number of collective MPI operations per MPI rank, coll_comm_time[s]: average time spent in collective MPI operations per MPI rank, delay_mpi[s]: total amount of MPI time spent in waiting caused by inefficient communication patterns, delay_mpi_ratio: ratio of waiting time caused by MPI to total time spent in MPI, time_omp[s]: total time spent in OpenMP parallel regions, delay_omp[s]: total amount of OpenMP synchronization overhead, delay_omp_ratio: ratio of synchronization overhead time in OpenMP to total time spent in OpenMP, memory_footprint[kB]: average memory footprint per MPI rank, cache_usage_intensity: ratio of total number of cache hits to the total number of cache accesses, IPC: ratio of total instructions executed to the total number of cycles, time_no_vec[s]: wall clock time without compiler vectorization, vec_eff: ratio of total wall time of the reference run to the total wall time without vectorization, time_no_fma: total wall time with disabled FMA, fma_eff: ratio of total wall time of the reference run to the total wall time without FMA



package. This avoids interference between different work package runs and creates a reproducible structure. For dependent work packages, symbolic links are created to the parent work package, for user access.

We added extra provenance tracking features to ParFlow simulation runs including converting unannotated ParFlow binary file model simulation output to a more portable NetCDF output containing standardized meta-data enrichment for data prove-
nance tracking using CMOR and CF standards and incorporation of all ParFlow model settings (Eaton et al., 2003). This was developed in accordance with state-of-the-art data lifecycle management and to maintain interoperability (Stodden et al., 2016).

In addition, the postprocess and analysis step we developed contains an archive process at the end of the modeling chain, which documents and collates the model input, model simulation scripts, model submission scripts, postprocessed output, and application code in such a fashion that the archived output can be downloaded and rerun following the instructions in the
simulation documentation, without need for any additional input, a practice recommended by Hutton et al. (2016).

## 2.2 Test case experimental design

In order to demonstrate the applicability of the RCF, a weak scaling demonstration study with an idealized overland flow test case was set up for ParFlow (Maxwell et al., 2015). ParFlow (v3.2, https://github.com/parflow) is a massively parallel, physics-based integrated watershed model, which simulates fully coupled, dynamic 2D/3D hydrological, groundwater and
land-surface processes suitable for large scale problems. ParFlow is used extensively in research on the water cycle in idealized and real data setups as part of process studies, forecasts, data assimilation experiments, hind-casts as well as regional climate change studies from the plot-scale to the continent, ranging from days to years. Saturated and variably saturated subsurface flow in heterogeneous porous media are simulated in three spatial dimensions using a Newton-Krylov nonlinear solver (Ashby and Falgout, 1996; Jones and Woodward, 2001; Maxwell, 2013) and multigrid preconditioners, where the three-dimensional
Richards equation is discretized based on cell centered finite differences. ParFlow also features coupled surface-subsurface flow which allows for hillslope runoff and channel routing (Kollet and Maxwell, 2006). Because it is fully coupled to the Common Land Model (CLM), a land surface model, ParFlow can incorporate exchange processes at the land surface including the effects of vegetation (Maxwell and Miller, 2005; Kollet and Maxwell, 2008). Other features include a parallel data assimilation scheme using the Parallel Data Assimilation Framework (PDAF) from Nerger and Hiller (2013), with an ensemble Kalman
filter, allowing observations to be ingested into the model to improve forecasts (Kurtz et al., 2016). An octree space partitioning algorithm is used to depict complex structures in three-dimensional space, such as topography, different hydrologic facies, and watershed boundaries. ParFlow parallel I/O is via task-local and shared files in a binary format for each time step. ParFlow is also part of fully coupled model systems such as the Terrestrial Systems Modeling Platform (TerrSysMP) (Shrestha et al., 2014) or PF.WRF (Maxwell et al., 2011), which can reproduce the water cycle from deep aquifers into the atmosphere.
A three-dimensional sinusoidal topography as shown in Figure 3 was used as the computational domain with a lateral spatial discretization of $\Delta x = \Delta y = 1$ m and a vertical grid spacing of $\Delta z = 0.5$ m; the grid size, n, was set to nx = ny = 50 and nz = 40 resulting in 100,000 unknowns per CPU core, with one MPI task per core. In order to simulate surface runoff from the high to the low topographic regions with subsequent water pooling and infiltration, a constant precipitation flux of 10 mm/hour was applied. The water table was implemented as a constant head boundary condition at the bottom of the domain



with an unsaturated zone above, 10 m below the land surface. The heterogeneous subsurface was simulated as a spatially uncorrelated, log-transformed Gaussian random field of the saturated hydraulic conductivity with a variance ranging over one order of magnitude. The soil porosity and permeability were set to 0.25 m/day.

The weak scaling experiment is defined as how the solution time varies with the number of processors for a fixed problem
size of 100,000 degrees of freedom per processor. The horizontal (nx,ny) grid size is increased but the number of cells in the vertical direction, nz, remain constant. All model configurations were run for 10 hours with a time step size of $\Delta t = 0.5$ hr.

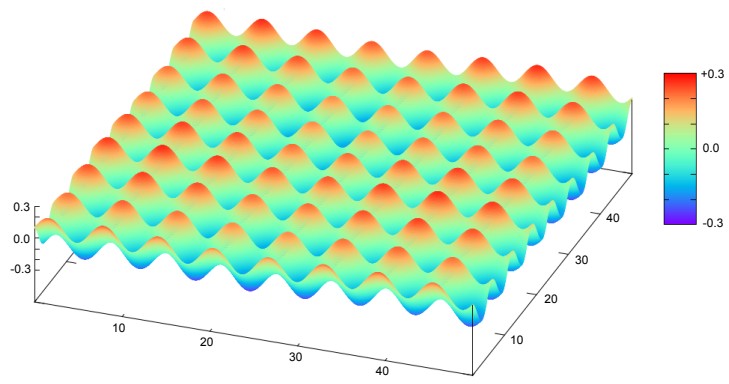

**Figure 3.** Model set up, showing cross-sectional domain and sinusoidal topography variation from the top of the model (z=20) for each processor

## 3 Profiling case study

In this section, we present the different steps and results of the benchmarking and profiling study, which demonstrate the usefulness of our RCF. For this study we use the highly scalable IBM Blue Gene/Q system JUQUEEN. JUQUEEN features a
total of 458 752 cores from 1 024 PowerPC A2 16-core, four-way simultaneous multithreading CPUs in each of the 28 racks and a total of 448 TB main memory with a Linpack performance of 5.0 Petaflops. ParFlow is running under Linux microkernels on compute nodes using IBM XL compilers and a proprietary MPI library and a GPFS filesystem. See appendix A for an overview of the RCF used for this case study, including excerpts of the XML files used.

### 3.1 Porting ParFlow to JUQUEEN

When porting ParFlow onto JUQUEEN, we found the optimal commonly used compilation flag which did not compromise accuracy with the IBM XL C compiler (v12.1) for ParFlow to be `-O3 -qhot -qarch=qp -qtune=qp` (see Table 1).

Accuracy was determined to six significant figures. This is in agreement with the results with the fully coupled TerrSysMP model system on JUQUEEN in Gasper et al. (2014). Using the `-O3` compilation flag resulted in a speedup of close to factor





of 2 when running with 16 MPI tasks on 16 CPU cores (one compute node on Blue Gene/Q, no multithreading). The timing results were compiled using JUBE's result parser functionality, which was run as a post processing step.

| IBM XL COMPILER FLAGS | TIME [S] |
|---|---|
| -O1 -qhot -qarch=qp -qtune=qp | 203 |
| -O2 -qhot -qarch=qp -qtune=qp | 203 |
| -O3 -qhot -qarch=qp -qtune=qp | 110 |

**Table 1.** Time taken to run the ParFlow test case on JUQUEEN (IBM Blue Gene/Q) with 16 MPI processes using three different commonly used compiler flag optimizations.

### 3.2 Profiling results

The following section describes the results from the demonstration scaling study, following the code performance "health check" protocol given in section 2.1.2.

**Analysis of time spent in ParFlow functions**

As a first step, a breakdown of the time spent in each annotated region of ParFlow was obtained via internal timings in ParFlow and a Score-P profile measurement (see Appendix B2.1 for Score-P profiling settings), visualized as a bar chart in Figure 4. From the breakdown it is clear that the core component of ParFlow is the computation of the solution to a system of nonlinear equations, reflected in Figure 4, where most of the wallclock time is spent in the blue regions which make up the time spent getting a solution via a nonlinear solve step. A large part of the nonlinear solver's workflow can be summarized in two steps, which are as follows: the initialization of the problem for the specific input and the actual solver loop. The last two steps reside in the `KINSol` routine which is a component of the SUNDIALS solver library (Hindmarsh et al., 2005). Therefore, the nonlinear solve routine and its components are the focus of interest for reducing ParFlow's runtime. The nonlinear solve loop performs the computational process of computing an approximate linear solution (`KINSpgmrSolve`) where the intermediate solution is updated every iteration until the desired convergence or tolerance is reached. Those two aforementioned steps within the nonlinear solve loop are manually annotated in the source code and we will focus on these for our results. For simplicity, we have shortened these two steps to `setup_solver` and `solver_loop` respectively and will refer to them by this nomenclature henceforth.

**Scalability**

Our scalability analysis of ParFlow is again based on the Score-P profile measurement. Figure 5 shows a plot of the execution time versus the number of MPI processes when running the weak scaling experiment as outlined in Section 2.2, broken down into the two regions of interest: `setup_solver` and `solver_loop`. The behavior of both regions show an increase of execution time with an increase in the number of processes, though the `setup_solver` region shows better performance in comparison to the `solver_loop`. However, the strong scaling efficiency profile is comparable to similar codes (Mills et al., 2007).





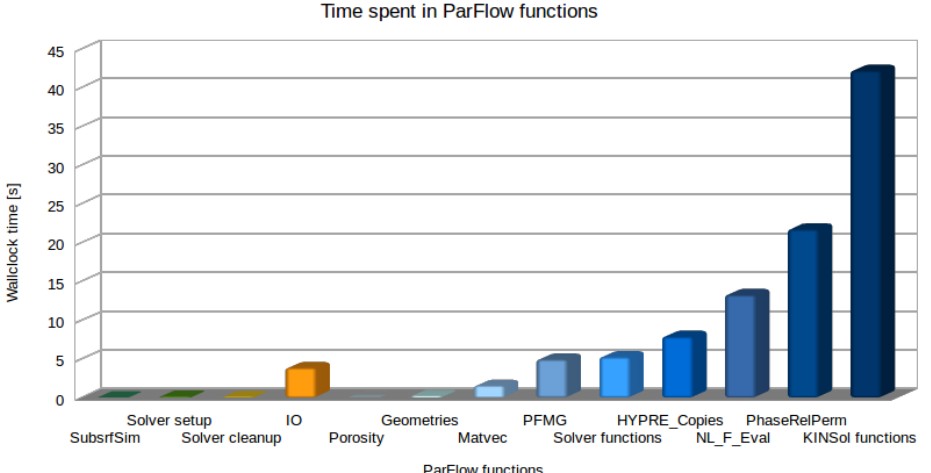

**Figure 4.** Time spent in ParFlow functions or routines, where the functions/routines can be divided into four categories, set up, clean up, I/O and solve. The functions in the category "set up" are depicted in green: SubsrfSim- setting up the domain, Solver setup- initializing the solver. The functions in the category "clean up" are depicted in yellow: Solver cleanup- finalizing the solver. The functions in the category "I/O" are depicted in orange: PFB I/O- ParFlow binary I/O. The functions in the category "solve" are depicted in blue: Porosity- calculation of the porosity matrix, Geometries- calculation of the simulation domain, MatVec- matrix and vector operations, PFMG:Geometric Multigrid Preconditioner from HYPRE, Solver functions:miscellaneous functions, HYPRE_Copies- copying data within HYPRE, NL_F_Eval- setting up the physics and field variables for the next iteration, PhaseRelPerm- setting up the permeability matrix, KINSol functions- nonlinear solver functions from SUNDIALS

At 32 768 MPI processes, the weak scaling efficiency $E_{ws}$, which is computed as a relation of the amount of time to complete a work unit with one process to the amount of time to complete N of the same work units with N processes (see Equation 1),

$$E_{ws} = \frac{T_1}{T_N} \tag{1}$$

drops to approximately 21%. In order characterize parallel applications, Rosas et al. (2014) developed auxiliary efficiency
5    metrics, i.e., load balance efficiency, communication efficiency, and parallel efficiency, which are described in more detail below. Using Cube's derived metric feature (Zhukov et al., 2015), we can derive these efficiency metrics from the Score-P profile data automatically.

Load balance efficiency, $E_{lb}$, is defined as a relation between average computation, $\overline{T}$, and maximal computation time, $T_{max}$,

10    $$E_{lb} = \frac{\overline{T}}{T_{max}} \tag{2}$$

Communication efficiency $E_{com}$ is defined as a relation between $T_{max}$ and total execution time, $T_{tot}$,

$$E_{com} = \frac{T_{max}}{T_{tot}} \tag{3}$$





Parallel efficiency, $E_{par}$, is defined as a product of load balance and communication efficiencies

$$E_{par} = E_{lb}E_{com} = \frac{\overline{T}}{T_{tot}} \tag{4}$$

Thus, for 32 768 MPI processes, the parallel efficiency drops to a reasonable value of 63% (Table 2). Further inspection of the breakdown between computation and communication (Figure 6) shows that total communication is considerably increasing at

scale, whereas `setup_solver` and `solver_loop` communication time do not contribute much to it. The slight increase in communication time in those two routines could be attributed to `MPI_Allreduce` in `setup_solver` and `MPI_Waitall` in `solver_loop` by further breaking down the communication routines (see Figure 7), however, the main scalability breaker is outside of those code sections, e.g., in the initialization phase or the preconditioner. This is an example where more in-depth analysis is needed after the initial health examination to clarify which part of the code must be improved.

| # MPI processes | Weak Scaling Efficiency | Load Balance Efficiency | Communication Efficiency | Parallel Efficiency [%] |
|---|---|---|---|---|
| 1024 | 100.0 | 96.33 | 96.84 | 93.29 |
| 2048 | 84.65 | 97.19 | 96.81 | 94.08 |
| 4096 | 83.50 | 97.15 | 81.95 | 79.61 |
| 8192 | 53.65 | 97.88 | 74.98 | 73.39 |
| 16384 | 50.07 | 98.01 | 68.71 | 67.34 |
| 32768 | 20.90 | 98.32 | 64.45 | 63.36 |

**Table 2.** Weak scaling efficiency, load balance efficiency, communication efficiency, and parallel efficiency, running the weak scaling experiment up to 32 768 MPI processes.

**Communication**

Time spent in communication grows with increasing number of processes (see Figures 6 and 7). For example, the communication time constitutes 37% of the total time when running the test case with 32 768 MPI processes. The main contributors to communication time within the regions of interest are `MPI_Allreduce` in `setup_solver` and `MPI_Waitall` in `solver_loop`. However, the main communication problem is outside of `setup_solver` and `solver_loop` at the ini-

tialization phase. A trace analysis using Scalasca identified a costly wait-state pattern constituting 23% of total time in the initialization phase occurring in `MPI_Allreduce` in the preconditioner of the HYPRE v2.10.1 library.

**Serial performance** A Score-P profile measurement with hardware performance counters was used to inspect serial performance. To describe serial performance, we use the instructions per cycle metric (IPC), i.e., the ratio of total instructions executed and the total number of cycles. The serial performance for the test case (Section 2.2) with 1 024 MPI processes shows

lower than ideal values of IPC. For example, `solver_loop` has an IPC value of 0.31 out of 2. Potential reasons for low IPC are pipeline stalls, cache misses, and mis-predicted branches (John and Rubio, 2008). Therefore, additional measurements with hardware counters were collected. Analysis of these counters show that a significant number of cache misses in L1, stalls, and mis-predicted branches occur in the following routines: `RichardsJacobianEval`, `PhaseRelPerm`, `Saturation`, and `NlFunctionEval`. Since JUQUEEN is based on an in-order instruction execution model, meaning instructions are fetched,



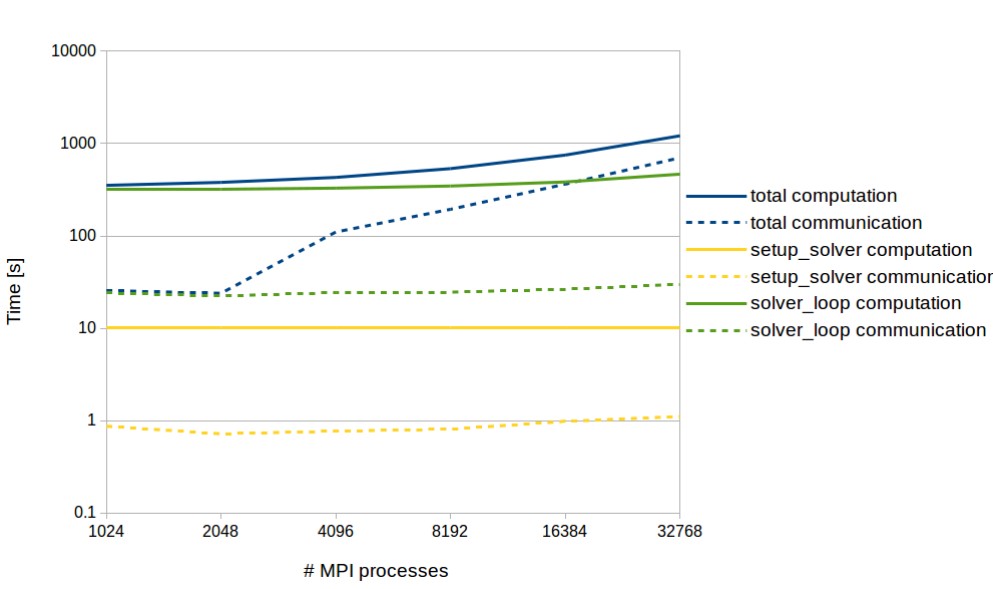

**Figure 5.** Execution time versus the number of MPI processes for the regions of interest, running the weak scaling experiment up to 32768 MPI processes.

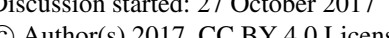

**Figure 6.** Wallclock time for communication versus computation for the regions of interest for the weak scaling experiment using the test case described in Section 2.2





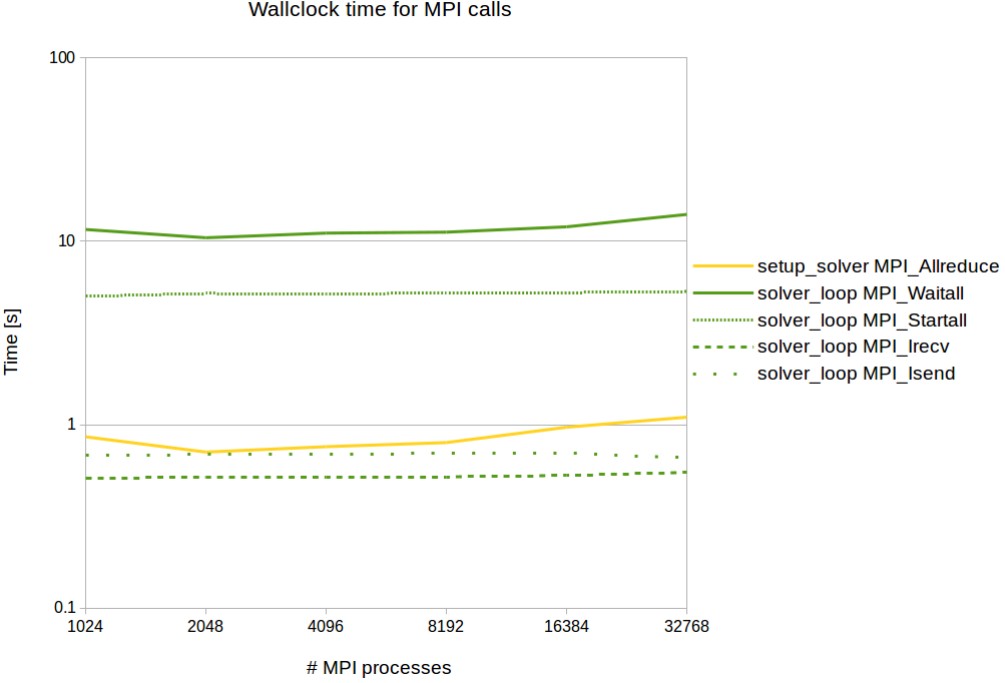

**Figure 7.** Wallclock time for MPI calls for the weak scaling experiment using the test case described in Section 2.2

executed, and committed in compiler-generated order, in case of an instruction stall, all ensuing instructions will stall as well. Branching on JUQUEEN is therefore very expensive and can cause pipeline stalls. Thus, the aforementioned routines may account for the low IPC values.

**Memory**

5    Using the idealized weak scaling test case described in Section 2.2, Score-P was used to track memory usage of the test case on JUQUEEN. Due to the idealized behaviour of the test case all MPI ranks needed roughly the same amount of memory. For example, at 1 024 processes, each rank needed roughly 95 MB and at 32 678 processes roughly 325 MB. Memory usage per MPI rank increases with scale as the entire grid information is redundantly stored on each MPI rank. This becomes a scalability breaker for ParFlow as we can see from Figure 8; there is a point at which the memory required will eclipse the memory avail-

10   able (at around 64 000 cores for this test case). In ParFlow, the most memory consuming routines are: `GetGridNeighbors`, `PFMGInitInstanceXtra`, `KinSolPC`, and `AllocateVectorData`.

## 3.3 Reproducibility

All simulation runs in the scaling study are separated into different subdirectories for each simulation run. All subdirectories include the XML scripts used by JUBE, the job submission scripts, the job logs, the model scripts, the postprocessing analysis





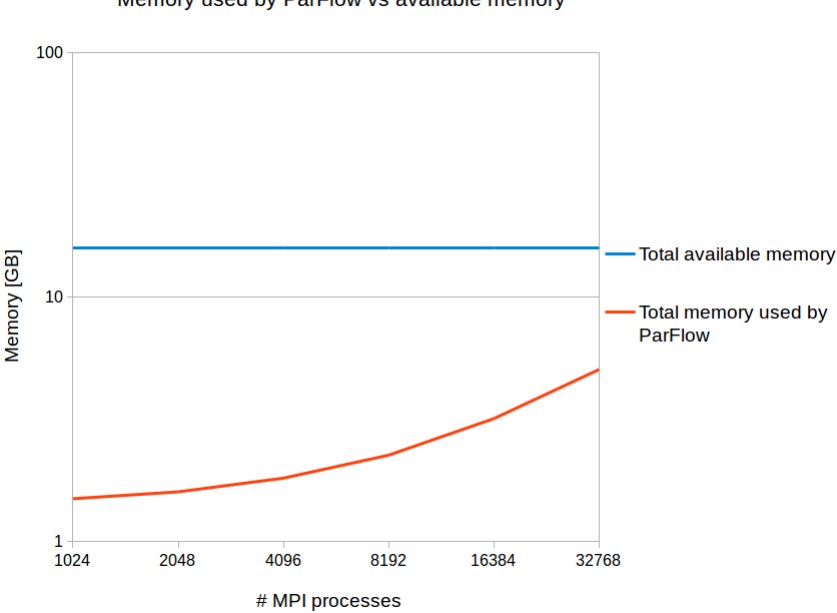

**Figure 8.** Memory usage of the weak scaling experiment described in Section 2.2 versus the total amount of memory available.

and a description of the version of ParFlow used along with the ParFlow binary itself. Each directory is self-contained such that the model can be rerun without using any other external tools or files. After the simulation is run and the postprocessing step has been executed, the directory is automatically archived for long term storage.

### 3.4 Outcomes of profiling case study

The detailed profiling work illuminated the main bottlenecks to scalability. These are memory use, time spent in communication, and time spent in acquiring the solution for each time step. In addition, for most geoscientific software including ParFlow, big data is also a barrier to scalability, with costly time spent processing data from the endian sensitive binary format to the highly portable NetCDF format along with time spent in data analysis where raw model output is processed to produce useful field variables. To tackle these barriers, the following development tasks have been initiated (Figure 9). To reduce the memory

used, to reduce the time spent in communication, and to address load imbalances due to inactive regions in model domains, Adaptive Mesh Refinement (AMR) is currently being implemented into ParFlow. As it stands, all cells store information about every other cell. This is reduced to neighboring cells under AMR which results in a decrease in memory use and a decrease in time spent in communication (Burstedde et al., 2017). To improve solution time and to enable use of heterogeneous architectures, different accelerator-enabled numerical libraries are being tested using our run harness, for a simplified version

of ParFlow, across different HPC architectures. To reduce time spent in preprocessing model input and postprocessing model output, a NetCDF reader and writer is under development. To reduce the amount of output needed to be written to disk, in-situ





visualization is under development, where raw output can be processed on-the-fly to useful field variables. There is still room for improvement with regards to serial performance. However to tackle this problem effectively, more in depth profiling is required before significant refactoring can occur. Naturally, we will use our run harness to then validate the effectiveness of these new developments and tuning efforts.

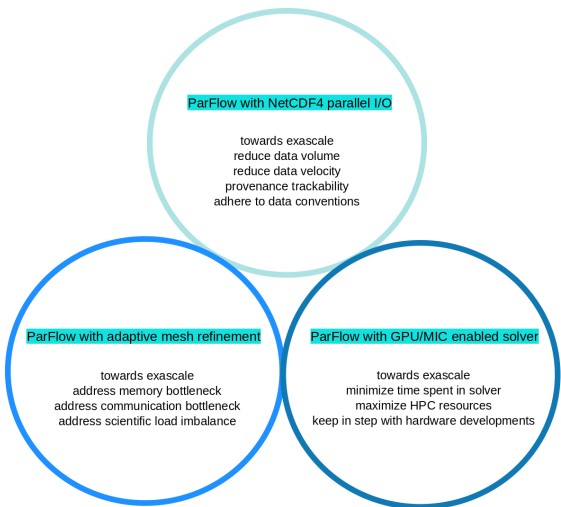

**Figure 9.** Outcome of the case study: Current developments addressing (i) portability and big data (ParFlow with NetCDF4 parallel I/O), (ii) memory bottleneck and scientific load balance (ParFlow with p4est), and (iii) next generation HPC architecture (ParFlow with GPU/MIC enabled solver)

## 4   Conclusions

Adapting to new developments in HPC architectures, software libraries, and infrastructures while ensuring reproducibility of simulation and analysis results has become challenging in the field of geoscience. Next generation massively parallel HPC systems require new coding paradigms and next generation geoscientific models are based on complex model implementations, and profiling, modeling and data processing workflows. Thus there is a strong need for a streamlined approach to model simulation runs, including profiling, porting, and provenance tracking.

In this article, we presented our streamlined approach, using JUBE as a workflow engine, so that a workflow can be executed and reproduced in a formalized way. We chose JUBE as it is lightweight and written in Python, and thus is easily portable to any supercomputer platform. Implementing a run control framework using a workflow engine for the complete modeling chain consisting of preprocessing, simulation run, and postprocessing leads to code that can be ported easily and tuned to any platform and combination of compilers including dependencies. Each simulation is automatically documented, accounting for provenance tracking, which leads to better supplementary code sharing and ultimately reproducibility. The relevant profiling toolset can be applied on any platform, leading to identification of bottlenecks, code tuning, refactoring, and ultimately more



efficient use of HPC resources. For example, the detailed profiling study of ParFlow led to the identification of bottlenecks and opportunities for code modernization (Figure 9) and significant improvement in all performance metrics.

The proposed approach helps the novice user as well as the developer and can be embedded into regression testing and a continuous integration approach. Using our RCF, testing, benchmarking, profiling, and running models is less time consuming and more robust, resulting in more efficient use of HPC resources, more strategic code development, and enhanced data integrity and reproducibility.

*Code and data availability.* The run control framework, data and version of ParFlow used in this paper are available for download via https://gitlab.maisondelasimulation.fr/EOCOE/Parflow upon request for access. A tarball containing the RCF, data and version of ParFlow relevant to this study has been included as supplementary information.

## Appendix A: Profiling tools implemented into the RCF

Table A1 describes the profiling tools which are currently supported by our RCF. New profiling tools can easily be added into the framework by adding to the `profiler.xml` file (See Appendix B2.1).

## Appendix B: Run control framework

Excerpts from the run control framework, used for the profiling study outlined in Section 3 are given in the following sections. The directory structure for the run harness used in the case study is shown in Figure B1. A python script, jubeRun.py combines the custom job specifications (`custom/weakScalingSinusoidal_Job_Juqueen.xml`), with the run control benchmark XML script (`driver/ParFlowRC_Benchmark.xml`) into one `execute.xml` file which can be parsed through JUBE, and then calls the JUBE run command with the newly created file as an argument. Machine specific profiling and job submission parameter sets are imported from XML structs given in the scripts `templates/platform.xml` and `templates/profiler.xml`, respectively, and the ParFlow model input parameter sets are imported from the structs given in `templates/ParFlow_model_input.xml`, based on the options specified in the custom job. All environment and submission scripts are stored in directories <machine>_files, all the profiling specific wrappers and filter files are stored in the directory profiler_data and all ParFlow model input is stored in the directory model (see Figure B1).

### B1 driver/ParFlowRC_Benchmark.xml

This script contains the steps to compile and run the benchmark suite, where the run step is dependent on the successful completion of the compilation step. Compilation parameters are set based on which HPC platform or machine the benchmark suite is run on and the profiling tool/s chosen. The user can specify the machine, the profiling tool/s, the ParFlow model, the domain size, the scaling parameters, and overwrite the default compilation and job submission parameters via the custom job XML script. The user can also describe an analysis step for the postprocessing of output.





| Performance Analysis Tool | Description |
|---|---|
| Score-P | Score-P is a community-maintained scalable instrumentation and performance measurement infrastructure for parallel codes. It can collect both profiles and event traces. <br> https://www.score-p.org |
| Scalasca | The Scalasca Trace Tools are a collection of scalable trace-based tools for in-depth analyses of concurrent behavior, in particular regarding wait states in communication and synchronization operations as well as their root causes. Supports Score-P traces since v2.0. <br> https://www.scalasca.org |
| Cube | Cube is an interactive analysis report explorer for Score-P profiles and Scalasca trace analyis reports. <br> http://www.scalasca.org/software/cube-4.x/ |
| Allinea Performance Report | Allinea Performance Report is a performance tool which provides a high-level overview of the runtime using a single-page report. <br> https://www.allinea.com/products/allinea-performance-reports |
| Extrae | Extrae is a measurement system which is able to collect traces for use with Paraver. <br> https://tools.bsc.es/extrae |
| Paraver | Paraver is a flexible and configurable performance analysis tool based on traces collected by the Extrae measurement system. It supports time-line views as well as histogram/statistics views on the trace data. <br> https://tools.bsc.es/paraver |
| Intel® Advisor | Advisor is a tool to analyze node-level performance issues, in particular regarding code vectorization and threading. <br> https://software.intel.com/en-us/intel-advisor-xe |
| Darshan | Darshan is tool to capture and characterize the I/O behavior of an application. <br> http://www.mcs.anl.gov/project/darshan-hpc-io-characterization-tool |
| PAPI | PAPI is a library providing a consistent interface for accessing hardware performance counters of CPUs and other components. While it can be called from application code directly, PAPI is more often used through other performance measurement systems such as Extrae and Score-P. <br> http://icl.utk.edu/papi |

**Table A1.** Description of of the performance analysis tools supported by our run harness infrastructure.





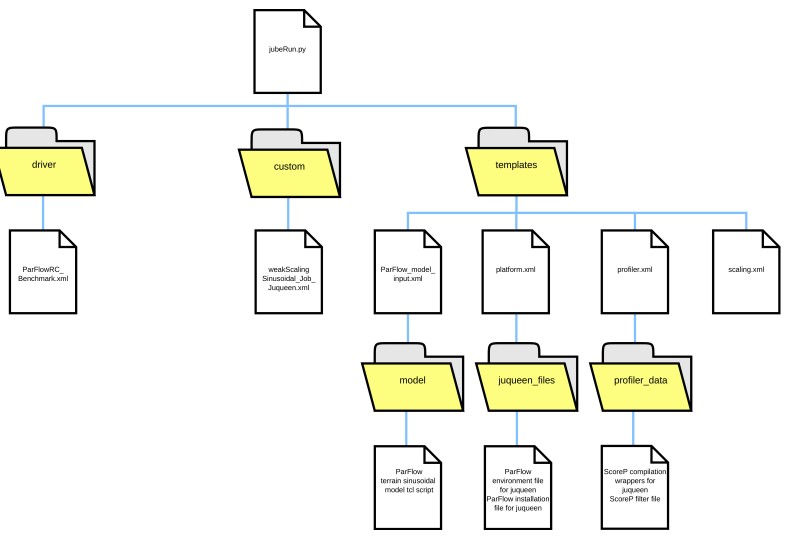

**Figure B1.** Directory structure of the run control framework used in the case study (Section 3)

```
   <?xml version="1.0" encoding="UTF-8"?>

   <jube>
    <!-- The purpose of this driver benchmark file is to facilitate the running of single ParFlow specific
        ↪ batch jobs with profiling in two steps: compile and run -->
 <!-- Either modify the benchmarkName, outputDir and runID here by hand or use the ./jubeRun.py -->
      <benchmark name="benchmarkName" outpath="outputDir/runID">
          <comment>
              ParFlow hydrological model testing and benchmarking (JUBE2)
          </comment>
          <!-- compile ParFlow -->
          <step name="setup_compile" export="true">
              <!-- To run: your own custom job parameters xml file should be put in by hand here- eg. custom/
                  ↪ my_custom_job.xml OR you can use the ./jubeRun.py and specify the name of your custom xml
↪  file as an argument -->
              <!-- The machine you are running on -->
              <use from="custom/customJob.xml">machineInfo</use>
              <!-- Your nominated profiler -->
              <use from="custom/customJob.xml">profiler</use>
<!-- Your nominated compilation parameters -->
              <use from="custom/customJob.xml">param_compile</use>
              <!-- Your nominated result -->
              <use from="custom/customJob.xml">resultInfo</use>
              <!-- End of custom input -->
```





```
          <use from="templates/profiler.xml:profiler_none_julia:profiler_none_juropa:profiler_none_jureca
              ↪ :profiler_none_juqueen:profiler_darshan_jureca:profiler_darshan_juqueen:
              ↪ profiler_scorep_jureca:profiler_scorep_juqueen:profiler_allinea_jureca:
              ↪ profiler_memusage_juqueen:profiler_extrae_jureca:profiler_extrae_juqueen:
              ↪ profiler_vectoradvise_jureca">
            profiler_${profiler}_${machine}
          </use>
          <use from="templates/profiler.xml:modules_none_julia:modules_none_juropa:modules_none_jureca:
              ↪ modules_none_juqueen:modules_darshan_jureca:modules_darshan_juqueen:modules_scorep_jureca
              ↪ :modules_scorep_juqueen:modules_allinea_jureca:modules_memusage_juqueen:
              ↪ modules_extrae_jureca:modules_extrae_juqueen:modules_vectoradvise_jureca">
            modules_${profiler}_${machine}
          </use>
          <use from="templates/platform.xml:files_compile_julia:files_compile_juropa:files_compile_jureca
              ↪ :files_compile_juqueen">files_compile_${machine}</use>
          <use from="templates/platform.xml:substitute_env_julia:substitute_env_juropa:
              ↪ substitute_env_jureca:substitute_env_juqueen">substitute_env_${machine}</use>
          <use from="templates/platform.xml:substitute_compile_julia:substitute_compile_juropa:
              ↪ substitute_compile_jureca:substitute_compile_juqueen">substitute_compile_${machine}</use>
          <do>time ./ParFlowInstall_${machine}.ksh</do>
      </step>
  <step name="setup_submit_run" depend="setup_compile" export="true">
          <!-- Your own custom benchmark setup files should be put in by hand here- see example: custom/
              ↪ weakScalingIdealised_job.xml OR you can use the ./juberun_script.sh to specify the name
              ↪ of your custom xml file as an argument -->
          <!-- The machine you are running on -->
          <use from="custom/customJob.xml">machineInfo</use>
          <!-- Your parflow parameters and files -->
          <use from="custom/customJob.xml">parflowInput</use>
          <!-- Your nominated profiler -->
          <use from="custom/customJob.xml">profiler</use>
          <!-- Your system parameters eg mail etc -->
          <use from="custom/customJob.xml">systemParameter</use>
          <!-- Your nominated scaling type and associated parameters -->
          <use from="custom/customJob.xml">param_domain</use>
          <!-- Your nominated batch job substitution parameters -->
          <use from="custom/customJob.xml">substituteBatch</use>
          <!-- End of custom input -->
           <use from="templates/ParFlow_model_input.xml:filesRun_sine:filesRun_europe:
              ↪ filesRun_europepminuset:filesRun_europeponding:filesRun_alpine:filesRun_LW">filesRun_${
              ↪ terrain}</use>
          <use from="templates/platform.xml">jobfiles</use>
          <use from="templates/profiler.xml:files_none_julia:files_none_juropa:files_none_jureca:
              ↪ files_none_juqueen:files_darshan_jureca:files_darshan_juqueen:files_scorep_jureca:
```





```
            ↪ files_scorep_juqueen:files_allinea_jureca:files_extrae_jureca:files_extrae_juqueen:
            ↪ files_vectoradvise_jureca">
            files_${profiler}_${machine}
        </use>
 <use from="templates/ParFlow_model_input.xml:substitute_maintcl_sine:substitute_maintcl_europe:
            ↪ substitute_maintcl_europepminuset:substitute_maintcl_europeponding:
            ↪ substitute_maintcl_alpine:substitute_maintcl_LW">substitute_maintcl_${terrain}</use>
        <use from="templates/ParFlow_model_input.xml:substitute_disttcl_sine:substitute_disttcl_europe:
            ↪ substitute_disttcl_europepminuset:substitute_disttcl_europeponding:
10          ↪ substitute_disttcl_alpine:substitute_disttcl_LW">substitute_disttcl_${terrain}</use>
        <use from="templates/ParFlow_model_input.xml:substitute_clm_sine:substitute_clm_europe:
            ↪ substitute_clm_europepminuset:substitute_clm_europeponding:substitute_clm_alpine:
            ↪ substitute_clm_LW">substitute_clm_${terrain}</use>
        <use from="templates/platform.xml:substitute_env_julia:substitute_env_juropa:
15          ↪ substitute_env_jureca:substitute_env_juqueen">substitute_env_${machine}</use>
        <use from="templates/platform.xml:filesRun_julia:filesRun_juropa:filesRun_jureca:
            ↪ filesRun_juqueen">filesRun_${machine}</use>
        <use from="templates/platform.xml:executeset_julia:executeset_juropa:executeset_jureca:
            ↪ executeset_juqueen">executeset_${machine}</use>
<do>source ./env_${machine}.ini</do>
        <!-- Your shell cmd -->
        <do>$shell_cmd</do>
        <!-- submit batch job -->
        <do done_file="$done_file">$submit $submit_script</do>
<do>$profilerpostprocess_step</do>
    </step>

    <!-- Your choice of result and analysis -->
    <analyser name="result_info">
<!-- Your nominated result -->
        <use from="custom/customJob.xml">result_pattern</use>
        <analyse step="setup_submit_run">
            <file>$result_file</file>
        </analyse>
</analyser>

    <result>
        <!-- Your nominated result -->
        <use>result_info</use>
<table name="measurement_result" style="csv" sort="tasks">
            <column>tasks</column>
            <column>taskspernode</column>
            <column>nodes</column>
            <column>process_result</column>
```



```
            </table>
        </result>

    </benchmark>
</jube>
```

## B2  templates

Template scripts are used in our run harness to capture default settings for HPC platform and model related parameters. Specific default settings can be overwritten when specified in the custom job XML script. In the case study, the template scripts used contain HPC platform related default settings for profiling tools, compilers, and job submission. In addition there are templates

for each type of ParFlow model, where the idealized overland flow model is called terrain.sine (See 2.2), and the type of scaling typically used for a benchmark suite, i.e., weak scaling and strong scaling.

### B2.1  templates/profiler.xml:scorep

This template XML script sets the default settings for each profiling tool where the following XML excerpt depicts the settings for the profiling tool Score-P on JUQUEEN.

```
<?xml version="1.0" encoding="UTF-8"?>
    <jube>
     <!-- ParFlow compilation flags for compilation with scorep -->
     <parameterset name="profiler_scorep_juqueen">
            $PARFLOW_COMPILE
20          $PARFLOW_EXTRACT_SRC
            $PARFLOW_DOWNLOAD_SRC
            "scorep --thread=none"

            <!-- install flags/substitution flags for scorep -->
25          ${jube_benchmark_home}/profiler_data/scorep/scorep_wrap</
                ↪ parameter>
            $path_to_wrapper/scorep_mpicc.sh
            $path_to_wrapper/scorep_mpicpp.sh
            $path_to_wrapper/scorep_mpif90.sh
            $starter
            
            <!-- this goes in submission script -->
            
source env_jureca.ini
    export SCOREP_TOTAL_MEMORY=500MB
    export SCOREP_METRIC_RUSAGE_PER_PROCESS=ru_inblock,ru_oublock
    export SCOREP_FILTERING_FILE=filter
```



```
    export SCAN_ANALYZE_OPTS="--verbose --time-correct"
    export SCOREP_METRIC_PAPI=PAPI_TOT_INS,PAPI_TOT_CYC,PAPI_L1_DCM,PAPI_L2_DCM
                
                
5                   $result_measurement scan $scorep_flag
                
                
                export SCOREP_WRAPPER=off
                
export SCOREP_WRAPPER=on
    export SCOREP_WRAPPER_INSTRUMENTER_FLAGS="--thread=none"
                
        </parameterset>
         <parameterset name="modules_scorep_juqueen">
    module load UNITE
    module load scorep
    module load scalasca
    module load papi
        </parameterset>
        <fileset name="files_scorep_juqueen">
                <copy>${jube_benchmark_home}/profiler_data/scorep/filter</copy>
        </fileset>
</jube>
```

## B2.2 templates/platform.xml:juqueen

This template XML script sets the default settings for each HPC platform where the following XML excerpt depicts the settings
for JUQUEEN.

```
    <?xml version="1.0" encoding="UTF-8"?>
<jube>
     <!-- default juqueen compilation parameters -->
     <parameterset name="param_compile_juqueen">
            
            <!-- directories -->
35          ${jube_benchmark_home}
            $PARFLOW_BASE/src
            r693
            $PARFLOW_SRC/parflow.693_juqueen
            <!-- control flags -->
40          1
            1
```

```
        1
        $PARFLOW_SRC
        0
        <!-- un-/distributed file output -->
5       1
        <!-- compile with parflow internal clm -->
        <!-- optimization flags -->
        -O3 -qhot -qarch=qp -qtune=qp
<!-- pftools compile flags -->
        'gcc'
        'gfortran'
        'gfortran'
        <!-- set patches premake etc to null (only used for profiling) -->
        
        
        
        
</parameterset>

     <!-- files needed for running jube on juqueen -->
        <fileset name="filesRun_juqueen">
        <copy>${jube_benchmark_home}/juqueen_files/env_juqueen.ini</copy>
<copy>${jube_benchmark_home}/funcs.py</copy>
     </fileset>

     <!-- files needed to compile ParFlow on juqueen -->
     <fileset name="files_compile_juqueen">
<copy>${jube_benchmark_home}/juqueen_files/ParFlowInstall_juqueen.ksh</copy>
        <copy>${jube_benchmark_home}/juqueen_files/env_juqueen.ini</copy>
     </fileset>

     <!-- substitution variables for compiler flags for juqueen -->
<substituteset name="substitute_compile_juqueen">
        <iofile in="ParFlowInstall_juqueen.ksh" out="ParFlowInstall_juqueen.ksh"/>
        <sub source="#PATHTOWRAPPER#" dest="$path_to_wrapper"/>
        <sub source="#WRAPPERFUNCTIONS#" dest="$wrapperfuncs"/>
        <sub source="#PATCHES#" dest="$patches"/>
<sub source="#PREMAKE#" dest="$premake"/>
        <sub source="#POSTMAKE#" dest="$postmake"/>
        <sub source="#OPT_FLAGS#" dest="$opt_flags"/>
        <sub source="#MPICC#" dest="$mpicc"/>
        <sub source="#MPIC++#" dest="$mpicpp"/>
```



```
            <sub source="#MPIF90#" dest="$mpif90"/>
            <sub source="#CC#" dest="$cc"/>
            <sub source="#F77#" dest="$f77"/>
            <sub source="#FC#" dest="$fc"/>
5       </substituteset>

        <!-- load the needed modules for the chosen profiler -->
        <substituteset name="substitute_env_juqueen">
            <iofile in="env_juqueen.ini" out="env_juqueen.ini"/>
10          <sub source="#MODULES#" dest="$module"/>
        </substituteset>

        <!-- default juqueen execution scripts and parameters -->
        <parameterset name="executeset_juqueen">
15          <!-- Jobscript handling -->
            ${jube_benchmark_home}/juqueen_files
            juqueen_submit.job
            runjob
            
20          ready
            <!-- Chainjob handling -->
            shared
            ${shared_folder}/jobid
            juqueen_ChainJob_Script.sh
25          false
        </parameterset>

        <!-- default juqueen job parameters -->
        <parameterset name="systemParameter_juqueen">
30          <!-- Default jobscript parameter -->
            llsubmit
            16
            16
            1
35          
            $nodes * $taskspernode
            
            
            $threadspertask
40          
            batch
            
            
            
```





```
            
            $jube_wp_envstr
            ALL
            job.out
5           job.err
            00:30:00
            
            
            
        </parameterset>

        <!-- substitution variables for the juqueen job submission script -->
        <substituteset name="executesub_juqueen">
<!-- Default jobscript substitution -->
        <iofile in="${submit_script}.in" out="$submit_script" />
        <sub source="#ENV#" dest="$env" />
        <sub source="#NOTIFY_EMAIL#" dest="$mail" />
        <sub source="#NOTIFICATION_TYPE#" dest="$notification" />
<sub source="#BENCHNAME#"
         dest="${jube_benchmark_name}_${jube_step_name}_${jube_wp_id}" />
        <sub source="#SIZE#" dest="$size" />
        <sub source="#RESID#" dest="$resID" />
        <sub source="#TASKS#" dest="$tasks" />
<sub source="#RANKSPERNODE#" dest="$taskspernode" />
        <sub source="#NTHREADS#" dest="$threadspertask" />
        <sub source="#TIME_LIMIT#" dest="$timelimit" />
        <sub source="#PREAMBLE#" dest="" />
        <sub source="#POSTAMBLE#" dest="" />
<sub source="#PREPROCESS#" dest="" />
        <sub source="#POSTPROCESS#" dest="" />
        <sub source="#QUEUE#" dest="$queue" />
        <sub source="#STARTER#" dest="$starter" />
        <sub source="#ARGS_STARTER#" dest="$args_starter" />
<sub source="#MEASUREMENT#" dest="" />
        <sub source="#RUNJOBENVS#" dest="$runjobenvs" />
        <sub source="#STDOUTLOGFILE#" dest="$outlogfile" />
        <sub source="#STDERRLOGFILE#" dest="$errlogfile" />
        <sub source="#EXECUTABLE#" dest="$executable" />
<sub source="#ARGS_EXECUTABLE#" dest="$args_exec" />
        <sub source="#FLAG#" dest="touch $done_file" />
        </substituteset>

        <!-- substitution variables for the juqueen chain job script -->
```





```
        <substituteset name="substitutechain_juqueen">
            <iofile in="${chainjobscript}.in" out="${chainjobscript}"/>
            <sub source="#NO_OF_JOBS#" dest="$no_of_jobs"/>
            <sub source="#JOBSCRIPT#" dest="$jobscript"/>
5       </substituteset>

</jube>
```

### B2.3 templates/ParFlow_model_input.xml

This template XML script sets the default settings for a series of ParFlow models set up over different domains. The following

10 XML excerpt depicts the settings for the idealized overland flow model, terrain.sine, used in the case study.

```
    <?xml version="1.0" encoding="UTF-8"?>
    <jube>
        <!-- files needed for the submit step for sinusoidal terrain-->
        <fileset name="filesRun_sine">
15          <copy name="$tclfile">${jube_benchmark_home}/model/$tclfile</copy>
        </fileset>

        <!-- substitution variables in the tcl for the sinusoidal terrain-->
        <substituteset name="substitute_maintcl_sine">
<iofile in="$tclfile" out="$tclfile"/>
            <sub source="#NX#" dest="$nx"/>
            <sub source="#NY#" dest="$ny"/>
            <sub source="#NZ#" dest="$nz"/>
            <sub source="#CALC_NX#" dest="$calcnx"/>
<sub source="#CALC_NY#" dest="$calcny"/>
            <sub source="#CALC_NZ#" dest="$calcnz"/>
            <sub source="#DX#" dest="$dx"/>
            <sub source="#DY#" dest="$dy"/>
            <sub source="#DZ#" dest="$dz"/>
<sub source="#TIMESTARTTIME#" dest="$TIMESTARTTIME"/>
            <sub source="#TIMESTOPTIME#" dest="$TIMESTOPTIME"/>
            <sub source="#TIMEDUMP#" dest="$TIMEDUMP"/>
            <sub source="#TIMEVALUE#" dest="$TIMEVALUE"/>
            <sub source="#SILO#" dest="$out_silo"/>
<sub source="#PFB#" dest="$out_pfb"/>
            <sub source="#NETCDF#" dest="$out_netcdf"/>
            <sub source="#STARTDATEYEAR#" dest="$startdateyear"/>
            <sub source="#STARTDATEMONTH#" dest="$startdatemonth"/>
            <sub source="#STARTDATEDAY#" dest="$startdateday"/>
</substituteset>
```

```
        <!-- substitution variables to distribute files on each processor for sinusoidal terrain -->
      <substituteset name="substitute_disttcl_sine">
      </substituteset>

            <!-- substitution variables to distribute clm files on each processor for sinusoidal terrain
               ↪  -->
      <substituteset name="substitute_clm_sine">
      </substituteset>

 </jube>
```

### B2.4    templates/scaling.xml

This template XML script sets the default domain settings for a given ParFlow model. The scaling parameters can be set such
that either one subdomain is spawned per thread (weak scaling) or such that the domain size does not change (strong scaling).

```
<?xml version="1.0" encoding="UTF-8"?>
<jube>
        <!-- Default weak scaling params -->
      <parameterset name="param_scalingWeak">
        100
        100
        50
        <!-- spawn one subdomain per thread -->
        $NX*$NP
        $NY*$NQ
        $NZ*$NR
      </parameterset>

      <!-- Default strong scaling params -->
      <parameterset name="param_scalingStrong">
        200
        150
        50
        <!-- the domain is absolute -->
        $NX
        $NY
        $NZ
      </parameterset>
 </jube>
```





## B3   custom/WeakScalingSinusoidal_Job_Juqueen.xml

The custom job script sets the HPC platform used, the profiling tool, the parflow model type, the domain extent and type of scaling and the postprocessing analysis step. The custom job script can also overwrite default compiler settings and job submission settings. This excerpt from the custom XML script used in the case study depicts the parameters chosen for the ParFlow model, the type of scaling, domain size and the job submission. All parameters which are comma separated such as tasks, in this excerpt, are parsed by JUBE as a parameter sweep so that each comma separated variable is iterated over to become a separate run.

```xml
<?xml version="1.0" encoding="UTF-8"?>
<jube>
    <!-- nominate the machine you are running on- eg. juqueen -->
    <parameterset name="machineInfo">
        juqueen
    </parameterset>

    <!-- Your choice of profiler -->
    <parameterset name="profiler">
      <!-- set the desired profiler -->
      scorep
    </parameterset>

    <!-- Set the desired tcl file and executive arguments -->
    <parameterset name="parflowInput">
      sine
      terrain.sine.tcl
      terrain.sine
      <!-- set the timing information -->
      0.0
      10.0
      1.0
      0.5
      <!-- set the output format -->
      False
      True
      False
      export NP=$(python -c "from funcs import
          ↪ decomp_dim_bq; decomp_dim_bq($tasks, $taskspernode, $nodes, 2, 1, 0)"); export NQ=$(
          ↪ python -c "from funcs import decomp_dim_bq; decomp_dim_bq($tasks, $taskspernode, $nodes,
          ↪ 2, 1, 1)"); tclsh $tclfile $NP $NQ
    </parameterset>

    <!-- Set compile options and the parameter dependencies -->
```



```
       <parameterset name="param_compile" init_with="templates/platform.xml:param_compile_juqueen">
         <!-- scaling type selection -->
         Weak
         <!-- file type selection -->
5        None
         <!-- Extraction params -->
         1
         1
         1
10        <!-- Parflow dir -->
         r693
         $PARFLOW_SRC/ParFlow.${PARFLOW_REV}_${machine}
             ↪ _clm_${profiler}
       </parameterset>

       <!-- Your choice of scaling parameters -->
        <parameterset name="param_domain" init_with="templates/scaling.xml:param_scalingWeak">
         50
         50
20        40
         1
         1
         0.5
       </parameterset>

       <!-- Custom job running parameters -->
        <parameterset name="systemParameter" init_with="templates/platform.xml:systemParameter_juqueen">
         <!-- reservation ID -->
30        
         <!-- User defined jobscript parameters -->
         PFLBM_profiler$profiler
         1024,2048,4096,8192,16384,32768,65536
         <!-- parameter name="tasks" type="int">1
35        32
         int(${tasks}/${taskspernode}+0.99999)</
             ↪ parameter>
         int(${tasks}/${taskspernode}+0.99999)</
             ↪ parameter>
40        00:30:00
         w.sharples@fz-juelich.de
         $$PARFLOW_DIR/pfsimulator/bin/parflow
         $exec_arg
         error
```





```
        </parameterset>

        <!-- setup the job run command and use platform.xml-->
        <substituteset name="substituteBatch" init_with="templates/platform.xml:executesub_juqueen">
5           <sub source="#RESID#" dest="$resID"/>
            <sub source="#BENCHNAME#" dest="$jobname"/>
            <sub source="#PREPROCESS#" dest="$preprocess"/>
            <sub source="#MEASUREMENT#" dest="$measurement"/>
            <sub source="#POSTPROCESS#" dest="$postprocess"/>
10          <sub source="#ARGS_STARTER#" dest="$args_starter_hard"/>
            <sub source="#STARTER#" dest="$profile_starter"/>
        </substituteset>

        <!-- Your choice of result and analysis -->
<parameterset name="resultInfo">
            <!-- set the desired measurement: eg. time, memory, etc -->
            time
            <!-- File/s to analyze for the result step for the Scaling BM -->
            job.err
</parameterset>

        <patternset name="result_pattern">
            <!-- set the pattern to look for from the output file/s -->
            <pattern name="process_time_min" type="int">${jube_pat_int}m</pattern>
25          <pattern name="process_time_sec" type="float">${jube_pat_fp}s</pattern>
            <pattern name="process_result" type="float" mode="python">${process_time_min}*60+${
              ↪ process_time_sec}</pattern>
        </patternset>

</jube>
```

*Competing interests.* The authors declare that there are no conflicts of interest.

*Acknowledgements.* The authors gratefully acknowledge the computing time granted by the JARA-HPC partition for the project: "Fractal
Scaling of Hydrodynamics at the Catchment Scale" on the supercomputer JUQUEEN at Forschungszentrum Jülich. This work was supported
by the Energy oriented Centre of Excellence (EoCoE), grant agreement number 76629, funded within the Horizon2020 framework of the
European Union and POP (676553), the Centre of Excellence Performance Optimisation and Productivity. In addition we acknowledge the
supercomputing support provided to us by the Simulation Laboratory Terrestrial Systems (www.hpsc-terrsys.de/simlab) of the Centre for
High Performance Scientific Computing in Terrestrial Systems (Geoverbund ABC/J) and the Jülich Supercomputing Centre (JSC).



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
