# Peer review of "A run control framework to streamline profiling, porting and tuning, simulation runs and provenance tracking of geoscientific applications"

_Geoscientific Model Development, 2017_

## Referee Comment (RC1) · Anonymous Referee #1 · 21 Nov 2017

Title: Best practice regarding the three P's: profiling, portability and provenance when running HPC geoscientific applications

Author(s): Wendy Sharples et al.

This paper presents a utility for controlling the execution and initial evaluation of an application (the ParFlow model) running in a (primarily) HPC framework. There are two levels to the framework described: the "run control framework" (or RCF) which itself utilises a more generic JUBE benchmark framework as a workflow engine. Essentially these provide a method of systematically defining, running and analysing some benchmarks - the authors also suggest it would be suitable for use for production simulations

as well.

The framework isolates multiple runs using parameters configured at set up time and keeps all the data produced along with a set of reports, allowing parameter sweeps with automated isolation of the various results (termed "provenance tracking" in the paper). A case study is presented utilising the framework to examine weak scaling (increasing the domain size) for the ParFlow model.

This paper is a difficult read, because there are three different strands within it: motivation, tooling, and the results of using the tools. They are well isolated by the sections, but each is somewhat unsatisfying on its own, and the links between are not as strong as I would like to see in a GMD journal paper. As it stands I do not think it is fit for full publication in GMD, but I think it could be made so with some reworking to make the material more accessible and relevant to the GMD audience.

There is some good material in the motivation, but it falls uncomfortably between being either a complete description of the portability, performance and reproducibility issues associated with geo-scientific modelling or an introduction to those elements for which the tools discussed later are either well suited or applied. It would be stronger if it were the latter.

The discussion of the tooling itself is incomplete in important details, yet full of detail (like the XML files in the appendix) which cannot be easily consumed by the reader because of a lack of appropriate explanation. There is no discussion of why this tool is any different from any other tool (e.g. what are the strengths and weaknesses with respect to the two workflow tools introduced in section 1)?

The results of the analysis of ParFlow are probably the strongest and most interesting parts of the paper, but because of the layout of the paper, introductory material such as the definitions of load balance, are mixed in with results and interpretation. I would rather this section had been organised to correspond to the (very useful) list of "health checks" which begins on the bottom of page 6. It might have been that the relevant

definitions (equations 1 to 4) could have appeared in section 2, since that's where these issues are first introduced. The results and scientific consequences could then be clearly identified in 3.4.

Specific Comments

1. I do not believe the title fairly reflects the material of the paper. The paper is not about best practice, it is primarily about one workflow/benchmarking application, although it does list elements of best practice and motivate some of the issues.

2. The paper begins with motivation with a selective list of references for how increased HPC might be used. The list is somewhat different from that usually presented which normally now includes increased use of data assimilation alongside increasing complexity, domain (spatial or temporal), resolution and ensemble size. It would be good to see this list inclusive of data assimilation and temporal extent and without quite so many references which don't add much value (there are so many it appears to be an attempt to be exhaustive, but it is clearly not exhaustive, better to have few or no exemplars than three or four each, because one is left wondering "why *these* ones"?).

3. There is then some material on the upcoming difficulties with performance portability which adds to the motivation, but the implication is that these are issues which can be solved by optimisation. In particular the paragraph beginning on line4 of page 3 begins by recognising the massive investments required to get performance, then implies that this investment can leverage analysis of existing codes using benchmarking tools such as the RCF/JUBE one discussed here. I think this section would be stronger if there was a clean separation between the aspirations of parameter sweeping and performance analysis, which is primarily about optimisation, and that of massive structural reorganisation of code such as was involved in Leutwyler et al. This is not to denigrate the importance of the former, but just to realistically recognise the scope. As written, the paper overstates it.

4. In the context of scope it would also be useful to identify where the tool might

have significant limitations, e.g, where it could interfere with other configuration and workflow managers (or be interfered with). This is not to suggest that the tool is not useful, or even powerful, for a particular class of problems; just that like all solutions, it almost certainly has limited applicability. It would be a service for potential users for the authors to provide some clarity on any known scope issues.

5. I think the paper confuses key issues around reproducibility. The implication of the discussions about reproducibility on page 3 and section 3.3 is that "if only the relevant parameters were documented, simulations would be reproducible". While this is undoubtedly *necessary*, it is far from *sufficient*, Baker et al. doi.org/10.5194/gmd-8-2829-2015 discuss the issue of ensuring that the science remains the same when hardware and software environments change. This paper would be stronger for identifying the distinction between these different issues of reproducibility and linking them to Irving's discussion and prior literature.

6. This might address the issue that there is only a cursory discussion of the issues associated with porting ParView to JUQUEEN - indeed, one might have expected the use of this framework to help with that process. "It was found that the optimal flags which did not compromise accuracy" with accuracy determined "to six figures". This reviewer has no idea what they meant by "accuracy" in this context, and the cursory argument suggests that important issues around solution stability were not investigated (despite the motivation being reproducibility).

7. In the discussion of the tools itself, the overall workflow is well described (Figure 1 etc and the excellent list provided for the "health examination"), but the discussion of the tool provides names of files, and then exemplar files (in XML, in the appendices). It's not clear at all how and why this framework is better than a bunch of scripts with input files - it would be considerably stronger if there was some discussion of how the tool exploits the XML files (is there a semantic structure inherent in the files beyond that provided by the use of XML to control syntax)? It might be that this is what the description JUBE reference provides, but I was unable to access the description of

JUBE hidden behind a paywall. Some kind of discussion about how the XML content links to JUBE actions would be helpful. In any case, I recommend removal of the XML files in the appendix, on their own they are inscrutable and provide little value. However, if they were provided in a repo with documentation as to function, they would provide useful complementary material.

8. The bulk of the case study shows the ability of the tools to generate information to understand the performance of the ParView model on this platform, and introduces some of the plans to alleviate performance bottlenecks (such as Adaptive Mesh Refinement). However, I did not fully understand the argument as to why this is the obvious next step from the current arrangement where all cells know about all other cells (why)?. This piece of the argument was another place where I felt that there was blurring together in this paper around issues of performance portability (optimising for a target architecture, but not changing the science), versus algorithmic improvements in performance (which involve changing the science).

9. Somewhere in the paper there needs to be some comparison to prior art and other similar tools which may address some part of the scope of these tools. (The description of GMD Development and Technical papers states: "Development and technical papers usually include a significant amount of evaluation against standard benchmarks, observations, and/or other model output as appropriate.") While, the key word may be *usually*, in the context of *this* paper I think there should be a section similar to the "Related Work" section that appears in many computer science and software engineering papers covering relevant strengths and weaknesses.

10. Finally, I note that the code is apparently available on request, but I think it will not get significant uptake without being both open source and having an open development process. Neither are required for GMD publication, I only mention these factors to encourage the authors to build a community around what looks like a useful tool for a certain class of problems.

---

## Referee Comment (RC2) · Anonymous Referee #2 · 28 Nov 2017

The paper presents the some of the challenges of geoscientific modelling on HPC resources, with emphasis on profiling and processing workflows. In order to address provenance, portability and profiling best practices, a run control framework (RCF) based on JUBE is described. The demonstration of the RCF is conducted in a weak scaling experiment in ParFlow, an integrated watershed model. Presentation of the tests results constitutes a notable - and interesting - part of the paper.

The paper is relatively easy to read in sections, but difficult to read as a whole. It covers several domains of expertise such as HPC, geoscientific modelling, software engineering, run harness and profiling. More work is required to present a coherent and uniformly detailed experiment in its full context. The profiling element is also substantially more detailed than the others to the point of overshadowing them. Furthermore, profiling results could be much better linked and/or mapped to the very informative "health check" section. As for layout of the "health check", typical diagnosis tools appended for each item does not convey the specific (overlapping or not) role or features of each software very well.

The authors list in section 3.4 the outcomes of the profiling case study. This entices the reader to believe that profiling will also advance provenance and portability. The article presents only a few elements of future work to support the full scope of the study, at least as the title describes it. It is also difficult for a reader to consider the tests results - the most detailed element of the paper - as sufficient proof of application of best practices in profiling, portability and provenance.

The test case experimental design for ParFlow reads like a completely disjoint section to the rest of the paper. It is only very lightly linked with previous or subsequent sections, for instance on the motivations behind that particular test case and how it leverages the RCF, the workflows or the HPC resources. It is unclear looking at the profiling results what are the implications for the test case or to ParFlow itself. This section offers a great potential to present ParFlow software and models (graphically), to show alternative test configurations or to contrast with real-world scenarios.

The paper mentions two other workflow engines (p.3 33) before rapidly shifting to JUBE (p.4 25). What are the advantages, limitations, similarities of JUBE compared to these other solutions? As is, considering that the results analysis of ParFlow provided by the RCF is a large (and interesting) part of the paper, a reader might be tempted to think that JUBE was selected first and pitted against other solutions a posteriori. It might be sufficient to cite major findings for the associated publication.

Specific Comments

1. Some sentences are too long (p.1 8-10, p.3 8-10, p.17 6-10). The messages conveyed by these long sentences is important for the coherence of the whole.

2. Figure 9 could be changed to a textual list without loss of content. Is there an overlap or interrelation between those developments? Is there development to do outside the scope of ParFlow, for instance in the run harness or workflows? What does "towards exascale" refers to exactly (cite or describe)? At what scale is the system operating at right now? What are the hurdles from terascale/petascale onto exascale that the presented work will limit or remove?

3. A very large part of the relevant information related to Figure 2 is in its legend. The reader might not care very much about the screen layout of the output. The reader might be interested in numerical values for each result on some occasions, but most values aren't described or introduced previously in the article. Most of all, the reader will most probably be interested in the metrics themselves and how they relate to profiling - or to some extend to portability or provenance, if applicable.

4. Section 3.1 is very short compared to others at this level, which diminishes its impact. It does not help with readability and flow. The section would benefit from an extension of concepts or reinforcement of links with other sections. It may also be merged elsewhere.

5. Some of the claims in the article are very lightly substantiated, insufficiently nuanced or lacking details. For instance (p.1 16 and p.19), the author claims that RCF is less time consuming and more robust, but less/more than what? Than without use of an RCF? The article concludes by claiming a more efficient use of HPC resources that was not clearly demonstrated; it reads like this is only implied because best practices were followed. If that is the case, it is suggested to better define and highlight these best practices.

6. The paper also mention in a few places costs notions ("invested effort", "paid off", "cost in resources") without providing any data or basic financial analysis. What was the approximate amount invested by articles cited? How much money or energy was

saved? This comment is provided without diminishing the (substantial) presented work or assuming that this information is essential.

7. The paper briefly mentions hybrid and heterogeneous architectures, but do not mention cloud computing. While a very different architecture than HPC, a reader might interested to see if any of the work presented can be applied in cloud computing environments (worfklows? packaged code? profiling? tools? models?). Commercial and scientific offers in HPC-as-a-service might prove an interesting option for the RCF. Absence of cloud computing discussion is not seen here as a limitation of the article, only a potential topic of interest.

8. There are also almost no mention of standards, except a brief sentence on (p.10 3-5). A reader might expect that such large scale systems with claims on portability and provenance do indeed follow standards instead of reinventing the wheel.

9. The code profiling section (p.6 6) makes it difficult to the reader to separate author contribution - by means of the RCF - to outputs from ScoreP and Scalasa. There is a long list of what software can "examine" those outputs. Why are those outputs compatible with all these software? Is it a standardized file format or structured data?

10. Alinea Performance Reports and Intel Vectorization Advisor are present in each item of the health examination, but both software weren't used. Still, they are recommended by the authors. Is the toolset of the experiment sufficient? What is the additional insight offered by Alinea and Intel's products that the other tools can't?

11. Interoperability is largely undefined throughout the text. There is only a single mention of interoperability (p.10 6) for "extra" features. Interoperability between what and what exactly? Was interoperability a critieria either in the conception of the test, the run harness or the workflow?

12. The subsequent paragraph - a very long sentence - mention download and rerun. Results on reruns would be welcomed if possible, either on JUQUEEN or better, on

other infrastructures.

13. The paper states that platform.xml can be easily extended or altered to include new systems. It is always easy to modify XML files, but not trivial to know what constitutes a valid modification and to successfully deploy it on other systems. Is there any tools to help a user, a developer, an administrator? Most discussion on portability revolves around XML files, compiler/linker flags and use of Python language. The paper concludes that the RCF using a workflow engine leads to code that can be ported easily. These conditions are important, but insufficient. A more thorough description of "environment preparation setup" might help a reader to better assess how close this particular run harness is compared to his own environment(s).The article would benefit from a better definition of portability. Ported from where to where? Any example of a second HPC infrastructure in your network? Precise what future work will advance portability. Other topics that could help a reader - this reviewer in particular - to assess portability could include virtual environments, software containers, software repositories and continuous integration frameworks.

14. Table 2 presents efficiencies measured during the weak scaling experiment. The authors states that more in-depth analysis is needed, but no strategy, best practices or future work is offered to the reader. Is this analysis to be conducted by a specialist, is it tool-assisted, what is the state of the art?

15. There is an assumption that the directory structure (Figure B1) "allows for run time provenance tracking" and "such that the model can be rerun without using any other external tools". The directory structure presented is most probably correct as a part of the RCF implementation. Still, it is unclear this is sufficient to insure provenance tracking or rerun. Is a tool set available to explore these directories and/or rerun models? Is it indexed in some form? Is there some other semantic information available that can be used?

---

## Author Comment (AC1) · 10 Jan 2018

We thank the anonymous reviewer for taking the time to read this paper thoroughly and providing the authors with constructive and thoughtful feedback. Addressing this feedback has greatly improved the paper. I have provided a pdf file of the text below also with our responses in green which might be easier to read.

Addressing Reviewer #1's concerns:

Title: Best practice regarding the three P's: profiling, portability and provenance when running HPC geoscientific applications Author(s): Wendy Sharples et al. This paper

presents a utility for controlling the execution and initial evaluation of an application (the ParFlow model) running in a (primarily) HPC framework. There are two levels to the framework described: the "run control framework" (or RCF) which itself utilises a more generic JUBE benchmark framework as a workflow engine. Essentially these provide a method of systematically defining, running and analysing some benchmarks - the authors also suggest it would be suitable for use for production simulations as well. The framework isolates multiple runs using parameters configured at set up time and keeps all the data produced along with a set of reports, allowing parameter sweeps with automated isolation of the various results (termed "provenance tracking" in the paper). A case study is presented utilising the framework to examine weak scaling (increasing the domain size) for the ParFlow model.

This paper is a difficult read, because there are three different strands within it: motivation, tooling, and the results of using the tools. They are well isolated by the sections, but each is somewhat unsatisfying on its own, and the links between are not as strong as I would like to see in a GMD journal paper. As it stands I do not think it is fit for full publication in GMD, but I think it could be made so with some reworking to make the material more accessible and relevant to the GMD audience.

There is some good material in the motivation, but it falls uncomfortably between being either a complete description of the portability, performance and reproducibility issues associated with geo-scientific modelling or an introduction to those elements for which the tools discussed later are either well suited or applied. It would be stronger if it were the latter.

>Our aim was to introduce those elements for which our RCF is a solution. We have therefore provided a better and more complete explanation of exactly what the RCF is, what its capabilities and functions are and how it is a best practice approach to aid portability, profiling and provenance tracking.

The discussion of the tooling itself is incomplete in important details, yet full of detail

(like the XML files in the appendix) which cannot be easily consumed by the reader because of a lack of appropriate explanation. There is no discussion of why this tool is any different from any other tool (e.g. what are the strengths and weaknesses with respect to the two workflow tools introduced in section 1)?

>This was an oversight on our part. We have now included a discussion of the relevant strengths and weaknesses of JUBE, ecFlow and cylc and have stated the reasons for choosing JUBE and that the RCF could be adapted for other workflow engines. We have also tried to better describe what the RCF is separate from the workflow engine used (JUBE) and how they are integrated with each other.

The results of the analysis of ParFlow are probably the strongest and most interesting parts of the paper, but because of the layout of the paper, introductory material such as the definitions of load balance, are mixed in with results and interpretation. I would rather this section had been organised to correspond to the (very useful) list of "health checks" which begins on the bottom of page 6. It might have been that the relevant definitions (equations 1 to 4) could have appeared in section 2, since that's where these issues are first introduced. The results and scientific consequences could then be clearly identified in 3.4.

>Thanks for this great suggestion. We have moved these definitions up to section 2, and mapped each health check step back to the results presented and have provided a more in depth description of the profiling tools used and the function of the RCF in automating the health check workflow.

Addressing the specific comments:

1. I do not believe the title fairly reflects the material of the paper. The paper is not about best practice, it is primarily about one workflow/benchmarking application, although it does list elements of best practice and motivate some of the issues.

We wanted to present the RCF as a best practice approach, but we accept that the

title does not totally reflect the material presented. We have updated the title to: A run control framework to streamline profiling, porting and tuning, simulation runs and provenance tracking of geoscientific applications

2. The paper begins with motivation with a selective list of references for how increased HPC might be used. The list is somewhat different from that usually presented which normally now includes increased use of data assimilation alongside increasing complexity, domain (spatial or temporal), resolution and ensemble size. It would be good to see this list inclusive of data assimilation and temporal extent and without quite so many references which don't add much value (there are so many it appears to be an attempt to be exhaustive, but it is clearly not exhaustive, better to have few or no exemplars than three or four each, because one is left wondering "why *these* ones"?).

This is a good point. We have updated the reference lists so that there are no more than 3 and e.g. is used more often to indicate the references are a subset of a broader scope of research (1st paragraph)

3. There is then some material on the upcoming difficulties with performance portability which adds to the motivation, but the implication is that these are issues which can be solved by optimisation. In particular the paragraph beginning on line4 of page 3 begins by recognising the massive investments required to get performance, then implies that this investment can leverage analysis of existing codes using benchmarking tools such as the RCF/JUBE one discussed here. I think this section would be stronger if there was a clean separation between the aspirations of parameter sweeping and performance analysis, which is primarily about optimisation, and that of massive structural reorganisation of code such as was involved in Leutwyler et al. This is not to denigrate the importance of the former, but just to realistically recognise the scope. As written, the paper overstates it.

Yes this is true. We meant to say that the RCF can streamline the investment as bottlenecks can be quickly identified because it has the relevant profiling tools integrated

within it to automate the performance engineering approach. We have tried to clarify/add to this explanation in the paragraph mentioned in page 3 and we refer back to this automation in section 3.

4. In the context of scope it would also be useful to identify where the tool might have significant limitations, e.g, where it could interfere with other configuration and workflow managers (or be interfered with). This is not to suggest that the tool is not useful, or even powerful, for a particular class of problems; just that like all solutions, it almost certainly has limited applicability. It would be a service for potential users for the authors to provide some clarity on any known scope issues.

Good suggestion. In the section on JUBE 2.1 paragraph 2 discussion of limitations have been added to and also what is common between JUBE, cylc and ecFlow is stated. In addition the advantages/disadvantages of each workflow engine is discussed along with an explanation of why we have chosen JUBE.

5. I think the paper confuses key issues around reproducibility. The implication of the discussions about reproducibility on page 3 and section 3.3 is that "if only the relevant parameters were documented, simulations would be reproducible". While this is undoubtedly *necessary*, it is far from *sufficient*, Baker et al. doi.org/10.5194/gmd-8-2829-2015 discuss the issue of ensuring that the science remains the same when hardware and software environments change. This paper would be stronger for identifying the distinction between these different issues of reproducibility and linking them to Irving's discussion and prior literature.

Our RCF actually goes much further than just documenting the relevant parameters. The whole simulation can be rerun as the code, the forcing data, the environment: source file with modules to be loaded, including a list of dependencies their versions, all extraneous scripts including job submission scripts and a complete model description are bundled together. If a user were to untar an output directory, they would be able to compile and rerun the experiment with JUBE using the RCF XML file in the directory,

on the same machine and obtain the same results. We have improved the description on page 3 and section 3.3. Also we have mentioned that there are some things beyond the users control such as different hardware and compilers when porting models and ways to ensure the science remains the same.

6. This might address the issue that there is only a cursory discussion of the issues associated with porting ParView to JUQUEEN - indeed, one might have expected the use of this framework to help with that process. "It was found that the optimal flags which did not compromise accuracy" with accuracy determined "to six figures". This reviewer has no idea what they meant by "accuracy" in this context, and the cursory argument suggests that important issues around solution stability were not investigated (despite the motivation being reproducibility).

Yes the test was not very well explained. It was briefly explained in section 2.3. We have expanded this section and made reference to this section in section 3.2.

7. In the discussion of the tools itself, the overall workflow is well described (Figure 1 etc and the excellent list provided for the "health examination"), but the discussion of the tool provides names of files, and then exemplar files (in XML, in the appendices). It's not clear at all how and why this framework is better than a bunch of scripts with input files - it would be considerably stronger if there was some discussion of how the tool exploits the XML files (is there a semantic structure inherent in the files beyond that provided by the use of XML to control syntax)? It might be that this is what the description JUBE reference provides, but I was unable to access the description of JUBE hidden behind a paywall. Some kind of discussion about how the XML content links to JUBE actions would be helpful. In any case, I recommend removal of the XML files in the appendix, on their own they are inscrutable and provide little value. However, if they were provided in a repo with documentation as to function, they would provide useful complementary material.

Good suggestion. We have moved the description of the RCF from the appendix to

section 2 to help provide a fuller understanding of exactly what it is. We have tried to emphasize the links between the XML content and JUBE actions in section 2.2 to explain how the RCF builds the overall configuration XML file to be run by JUBE. The xml files are provided in a tarred up directory submitted with this manuscript. There are readme files plus documentation in each script as to what it's functionality is. To that end we have removed the xml files from the appendix.

8. The bulk of the case study shows the ability of the tools to generate information to understand the performance of the ParView model on this platform, and introduces some of the plans to alleviate performance bottlenecks (such as Adaptive Mesh Refinement). However, I did not fully understand the argument as to why this is the obvious next step from the current arrangement where all cells know about all other cells (why)?. This piece of the argument was another place where I felt that there was blurring together in this paper around issues of performance portability (optimising for a target architecture, but not changing the science), versus algorithmic improvements in performance (which involve changing the science).

We agree that the integration with p4est was not very clear. This is a case of an algorithmic improvement which does not change the science. ParFlow coupled to p4est means that p4est is now the parallel mesh manager, currently focusing on uniform meshes. The approach was minimally invasive and preserves most of ParFlow's data structures, the configuration system and the setup and solver pipeline. We have now explained this in section 3.4 and also mentioned the state of the current mesh manager in section 3.3.

9. Somewhere in the paper there needs to be some comparison to prior art and other similar tools which may address some part of the scope of these tools. (The description of GMD Development and Technical papers states: "Development and technical papers usually include a significant amount of evaluation against standard benchmarks, observations, and/or other model output as appropriate.") While, the key word may be *usually*, in the context of *this* paper I think there should be a section similar to the

"RelatedWork" section that appears in many computer science and software engineering papers covering relevant strengths and weaknesses.

There is related work in terms of scaling and profiling ParFlow, which has been reported in references Kollet et al. 2010 and Gasper et al. 2014 for example. There is a report on comparisons between workflow engines cylc and ecFlow in Manubens-Gil 2016 which is now mentioned in our manuscript. Our reasons for choosing JUBE over cylc and ecFlow are now outlined clearly in the manuscript, where the main reason for choosing JUBE is that it is the workflow engine most compatible for use on JSC machines. However the RCF itself is a new development to streamline scientific software profiling, porting and tuning, running production runs and to aid reproducibility. To date, there are no "standard" benchmarks as such.

10. Finally, I note that the code is apparently available on request, but I think it will not get significant uptake without being both open source and having an open development process. Neither are required for GMD publication, I only mention these factors to encourage the authors to build a community around what looks like a useful tool for a certain class of problems.

A tarball has been included with the manuscript which the users can use with JUBE and any version of ParFlow with the test case described in the paper straight away. This should be enough to get a new user familiarized with the RCF enough to add to it for their own purpose. The reason why the whole RCF code is not open source is because the real world models included are developed by different scientists, many of which have not yet been published.

Please also note the supplement to this comment:
https://www.geosci-model-dev-discuss.net/gmd-2017-242/gmd-2017-242-AC1-supplement.pdf

---

## Author Comment (AC2) · 10 Jan 2018

We thank the anonymous reviewer for taking the time to read this paper thoroughly and providing the authors with constructive and thoughtful feedback. Addressing this feedback has greatly improved the paper. I have also included a pdf file with our responses in green which might be easier to read.

Addressing reviewer 2's concerns:

The paper presents the some of the challenges of geoscientific modelling on HPC resources, with emphasis on profiling and processing workflows. In order to address

provenance, portability and profiling best practices, a run control framework (RCF) based on JUBE is described. The demonstration of the RCF is conducted in a weak scaling experiment in ParFlow, an integrated watershed model. Presentation of the tests results constitutes a notable - and interesting - part of the paper.

The paper is relatively easy to read in sections, but difficult to read as a whole. It covers several domains of expertise such as HPC, geoscientific modelling, software engineering, run harness and profiling. More work is required to present a coherent and uniformly detailed experiment in its full context. The profiling element is also substantially more detailed than the others to the point of overshadowing them. Furthermore, profiling results could be much better linked and/or mapped to the very informative "health check" section. As for layout of the "health check", typical diagnosis tools appended for each item does not convey the specific (overlapping or not) role or features of each software very well.

>The suggestion to map the results back to the health check is a good one. We have also moved definitions in section 3 up to section 2 to the health check, and mapped each health check step back to the results presented. We have provided a more in depth description of the profiling tools used and their overlap. In addition we have detailed the function of the RCF in automating the health check workflow to try and provide a coherent experiment with which to demonstrate the advantages of using the RCF.

The authors list in section 3.4 the outcomes of the profiling case study. This entices the reader to believe that profiling will also advance provenance and portability. The article presents only a few elements of future work to support the full scope of the study, at least as the title describes it. It is also difficult for a reader to consider the tests results - the most detailed element of the paper - as sufficient proof of application of best practices in profiling, portability and provenance.

>Yes we can see how this might cause some confusion. We have deleted Figure 9 and

also the text to surrounding the NetCDF reader/writer to limit confusion for the reader. We have also clarified that the RCF is a best practice approach to profiling, portability and provenance by altering the title and also providing a better explanation of what the RCF is and how it is integrated with the workflow engine, JUBE. We have updated the title to: A run control framework to streamline profiling, porting and tuning, simulation runs and provenance tracking of geoscientific applications

The test case experimental design for ParFlow reads like a completely disjoint section to the rest of the paper. It is only very lightly linked with previous or subsequent sections, for instance on the motivations behind that particular test case and how it leverages the RCF, the workflows or the HPC resources. It is unclear looking at the profiling results what are the implications for the test case or to ParFlow itself. This section offers a great potential to present ParFlow software and models (graphically), to show alternative test configurations or to contrast with real-world scenarios.

>Thanks for these suggestions. We have now included the motivations for using this particular test case in order to illuminate bottlenecks using the health check procedure and the reasons why a "real world" test case might obscure certain results such as load balance. We have also shifted the test case model description to section 3 for better continuity.

The paper mentions two other workflow engines (p.3 33) before rapidly shifting to JUBE (p.4 25). What are the advantages, limitations, similarities of JUBE compared to these other solutions? As is, considering that the results analysis of ParFlow provided by the RCF is a large (and interesting) part of the paper, a reader might be tempted to think that JUBE was selected first and pitted against other solutions a posteriori. It might be sufficient to cite major findings for the associated publication.

>This was an oversight. We have now included a discussion of the relevant strengths and weaknesses of JUBE, ecFlow and cylc and have stated the reasons for choosing JUBE but have also made it clear that the RCF could be adapted for other workflow

engines and they share a lot of the same functionality.

Address specific comments

Specific Comments 1. Some sentences are too long (p.1 8-10, p.3 8-10, p.17 6-10). The messages con- veyed by these long sentences is important for the coherence of the whole.

Agreed- we have fixed these sentences.

2. Figure 9 could be changed to a textual list without loss of content. Is there an overlap or interrelation between those developments? Is there development to do outside the scope of ParFlow, for instance in the run harness or workflows? What does "towards exascale" refers to exactly (cite or describe)? At what scale is the system operating at right now? What are the hurdles from terascale/petascale onto exascale that the presented work will limit or remove?

We have followed your suggestion and removed Figure 9. From the profiling results it can be seen that due to memory required being greater than memory available, at around 64000 cores, ParFlow as it stands is not approaching exascale. Exascale com- puting refers to computing systems capable of at least one exaFLOPS, or a billion bil- lion calculations per second. ParFlow coupled with p4est as the parallel mesh handler, allows ParFlow to scale to the whole Juqueen machine (petascale). We have made this clearer in section 3 and introduced the exascale concept in the same section.

3. A very large part of the relevant information related to Figure 2 is in its legend. The reader might not care very much about the screen layout of the output. The reader might be interested in numerical values for each result on some occasions, but most values aren't described or introduced previously in the article. Most of all, the reader will most probably be interested in the metrics themselves and how they relate to profiling - or to some extend to portability or provenance, if applicable.

This is a good idea. The reader would benefit from knowing what each metric is used
for. We have tried to explain what each metric means and why it might be useful in table format in appendix A2 and thus reduced the length of the figure caption.

4. Section 3.1 is very short compared to others at this level, which diminishes its impact. It does not help with readability and flow. The section would benefit from an extension of concepts or reinforcement of links with other sections. It may also be merged elsewhere.

We have altered section 2 to include a more thorough description of the accuracy test we used to test compilation flags. We have expanded section 3.1 to include a reference to this test as well as a discussion of the relevant compilation flags used for the specific IBM XL and why we focused on the two specific flags as opposed to others available.

5. Some of the claims in the article are very lightly substantiated, insufficiently nuanced or lacking details. For instance (p.1 16 and p.19), the author claims that RCF is less time consuming and more robust, but less/more than what? Than without use of an RCF? The article concludes by claiming a more efficient use of HPC resources that was not clearly demonstrated; it reads like this is only implied because best practices were followed. If that is the case, it is suggested to better define and highlight these best practices.

Yes less time consuming than running by hand or developing a series of specialized scripts which only work for one model without integrated profiling tools. We have tried to make this clearer in all sections of the manuscript where this is mentioned and have updated the conclusions accordingly.

6. The paper also mention in a few places costs notions ("invested effort", "paid off", "cost in resources") without providing any data or basic financial analysis. What was the approximate amount invested by articles cited? How much money or energy was saved? This comment is provided without diminishing the (substantial) presented work or assuming that this information is essential.

We are assuming you mean the references mentioned in page 3, 2nd paragraph and have updated this to include an example (Leutwyler et al. 2016) of speedup values (∼3.6) and also the invested effort required (a team of developers)

7. The paper briefly mentions hybrid and heterogeneous architectures, but do not mention cloud computing. While a very different architecture than HPC, a reader might interested to see if any of the work presented can be applied in cloud computing environments (worfklows? packaged code? profiling? tools? models?). Commercial and scientific offers in HPC-as-a-service might prove an interesting option for the RCF. Absence of cloud computing discussion is not seen here as a limitation of the article, only a potential topic of interest.

Thanks for pointing this out. Yes the RCF could be adapted for cloud computing or a web based interface. However this would be a substantial effort for a technology that is as yet severely limited by bandwidth such that it performs substantially worse than just using one HPC system. So we think this is out of scope for what is currently available to HPC users.

8. There are also almost no mention of standards, except a brief sentence on (p.10 3-5). A reader might expect that such large scale systems with claims on portability and provenance do indeed follow standards instead of reinventing the wheel.

A brief mention of these standards without explanation was because the authors assumed that the readers would be familiar what standards we follow (CF and CMOR Metadata Conventions) in the section on provenance tracking. We've now expanded this section to include a discussion to describe the CF conventions and their purpose. We have also now added to this section a discussion on how Irving 2016's minimum standard points 1-4 are covered by the RCF.

9. The code profiling section (p.6 6) makes it difficult to the reader to separate author contribution - by means of the RCF - to outputs from ScoreP and Scalasa. There is a long list of what software can "examine" those outputs. Why are those outputs

compatible with all these software? Is it a standardized file format or structured data?

This is true. We have tried to make this clearer by explaining what ScoreP, Scalasca and Cube are in the text (rather than in the appendix) and to clarify what RCF does (collect and collate the performance metrics by parsing the various reports generated by the tools). There is a standard format for ScoreP and Scalasca output which Cube can parse and read.

10. Alinea Performance Reports and Intel Vectorization Advisor are present in each item of the health examination, but both software weren't used. Still, they are recommended by the authors. Is the toolset of the experiment sufficient? What is the additional insight offered by Alinea and Intel's products that the other tools can't?

The toolset used was determined by what was available on JUQUEEN. We've now explained this in section 2. There is a lot of overlap with different tools and we have now explained this in the same section. As to what tool one uses it depends on 1. what is available on a given machine and 2. personal preference. For example, Alinea products offer a nice way of presenting coarse-grained metrics for a novice user but do not offer more features than many other tools. And Intel vector advisor additionally can provide some more guidance than other software when it comes to identifying potential loops to be vectorized but this is not part of the initial health check guidelines and so these results are not mentioned.

11. Interoperability is largely undefined throughout the text. There is only a single mention of interoperability (p.10 6) for "extra" features. Interoperability between what and what exactly? Was interoperability a critieria either in the conception of the test, the run harness or the workflow?

What the authors meant by interoperability and did not explain very well, is that CMORized NetCDF files can be used on different architecture (big and little endian) and for different software (for use in various terrestrial systems software and visualization software). In the section we've added about CF conventions we have also

mentioned how this facilitates interoperability and what we mean by interoperability in this instance.

12. The subsequent paragraph - a very long sentence - mention download and rerun. Results on reruns would be welcomed if possible, either on JUQUEEN or better, on other infrastructures.

Yes, the profiling results obtained are the result of running the benchmark described 10 times to get an average and make sure the benchmark is running as it should- i.e. there should not be a huge variance between results. And we have now added this explanation to section 3. The authors think that showing how this benchmark performs on other architecture is really beyond the scope of the paper as we are primarily discussing the RCF and how it aids portability, profiling and provenance tracking. Of course discussion of the portability aspect should include what other machines the RCF is running on (good suggestion) so we have mentioned that this RCF is also being run on Julia (a prototype KNL cluster at JSC), Jureca (a general purpose cluster at JSC) and Juropa3 (a prototype testbed system at JSC) in the description of the RCF.

13. The paper states that platform.xml can be easily extended or altered to include new systems. It is always easy to modify XML files, but not trivial to know what constitutes a valid modification and to successfully deploy it on other systems. Is there any tools to help a user, a developer, an administrator? Most discussion on portability revolves around XML files, compiler/linker flags and use of Python language. The paper concludes that the RCF using a workflow engine leads to code that can be ported easily. These conditions are important, but insufficient. A more thorough description of "environment preparation setup" might help a reader to better assess how close this particular run harness is compared to his own environment(s).The article would benefit from a better definition of portability. Ported from where to where? Any example of a second HPC infrastructure in your network? Precise what future work will advance portability. Other topics that could help a reader - this reviewer in particular - to assess portability could include virtual environments, software containers, software

repositories and continuous integration frameworks.

This is a valid point. The RCF facilitates the environment set up through the use of loading modules, most if not all HPC systems contain this feature, thus this is one of the reasons why the RCF is portable- we have now mentioned this in section 2. Within the platform XML file, there are structs defined with standard parameters for run time arguments and compilation flags which can then be used for any HPC system with particular versions of compilers and profiling tools. We have added this to the description in section 2.3 (code portability).

14. Table 2 presents efficiencies measured during the weak scaling experiment. The authors states that more in-depth analysis is needed, but no strategy, best practices or future work is offered to the reader. Is this analysis to be conducted by a specialist, is it tool-assisted, what is the state of the art?

Good point. More in depth profiling would be much more effective together with performance analysis engineers. We have added this explanation and have put in an example of future work that we are currently conducting with the aid of performance analysis specialists: vectorization of individual loops.

15. There is an assumption that the directory structure (Figure B1) "allows for run time provenance tracking" and "such that the model can be rerun without using any other external tools". The directory structure presented is most probably correct as a part of the RCF implementation. Still, it is unclear this is sufficient to insure provenance tracking or rerun. Is a tool set available to explore these directories and/or rerun models? Is it indexed in some form? Is there some other semantic information available that can be used?

What is meant is that whole simulation can be rerun as the code, the forcing data, the environment: modules loaded, including a list of dependencies their versions, all extraneous scripts including job submission scripts and a complete model description are bundled together in one output directory. If a user were to untar an output directory,
they would be able to compile and rerun the experiment using the XML file contained in the directory, with JUBE, on the same machine and obtain the same results. We have tried to improve the description on page 3 and in section 3. Also we have mentioned that there are additional features beyond the users control such as different hardware when porting models and ways to ensure the scientific results remain the same.

Please also note the supplement to this comment:
https://www.geosci-model-dev-discuss.net/gmd-2017-242/gmd-2017-242-AC2-supplement.pdf
* * *

---

## Referee Report (RR1)

| Title | Author(s) | MS no. |
| --- | --- | --- |
| A run control framework to streamline profiling, porting and tuning, simulation runs and provenance tracking of geoscientific applications | Wendy Sharples et al. | gmd-2017-242 (resubmission) |

This paper presents a utility for controlling the execution and initial evaluation of an application (the ParFlow model) running in a (primarily) HPC framework. There are two levels to the framework described: the "run control framework" (or RCF) which itself utilises a more generic JUBE benchmark framework as a workflow engine. Essentially these provide a method of systematically defining, running and analysing some benchmarks - the authors also suggest it would be suitable for use for production simulations as well.

The paper is a heavily modified resubmission, and was previously titled: "Best practice regarding the three P's: profiling, portability and provenance when running HPC geoscientific applications".

The authors have made considerable efforts to respond to the previous reviews, and the paper is much improved, however there are still a number of issues. I think it could appear in GMD provided these issues are addressed to the satisfaction of the editor.

Most of my issues now reside in the front material, the body of the paper describing the use of the tool with paraview is much improved, and I am happy for that part to appear more or less as is.

**Major Issues**

1. The goal of the paper is still not clear - but only because it is still obscured by what still feels like an excessive emphasis on motivation. I would ask the authors the following question: "If you read the abstract, what would you expect to find in the body of the paper?" Half the abstract is motivation, which seems wrong. Results of using this tool with ParFlow are not even mentioned. Along the same lines, it is page 4 before the introduction gets round to telling us the bulk of the paper is about the RCF.

   - These issues could relatively easily be addressed by reworking the abstract and either removing much of the existing introduction, or moving much of it to a motivation section immediately following.

2. It is good to see there is now a discussion of other tools in section 2.1, but the material is not well connected, misses the point in a number of crucial ways, and (I would assert) wrong in some of the statements. There is no pre-existing taxonomy of tools in this space, and it would be unreasonable to expect the authors to have real experience with these tools, so getting the level of discussion right is not trivial, but

   1. Page 5, line 9 mashes a description of JUBE into a description of other tools. At the very least this is a new paragraph.

2. It is not clear why all the emphasis on XML. All XML provides is a syntax (which is obviously useful) but the statement that cylc has "its own scripting" (line 15) is mixing action (scripting) with the syntax definition (XML). Cylc actually uses INI and Jinja2 for syntax, what is interesting about the differences is not whether one uses XML or INI, but what semantics exist in the configuration. What can they do? 3. The discussion about the platform constraints on submission belongs in its own paragraph (but I would ask why, with 2 hour limits, they can't run cylc - or any other tool - on a third system and simply poll through the login nodes using ssh tunnelling).

3. No one expects such a comparison to assert that cylc or any other tool is *better or worse* than their tools, simply stating different capabilities is all that is really required. Clealy the JUBE RCF roadmap will differ from those tools, and it would be fine for them to describe and build other tools, even if they had the same or poorer functionalty - which is clearly not the case, there are some real advantages to describe here! The problem is that some of those advantages become clearer once the use case is fully described. It might be that this comparison could go in a paragraph preceding the conclusions, but whatever is done, all tools have strengths and weaknesses, this paragraph currently reads like a "defense" rather than a "comparison". 5. (Some of those advantages are actually in the semantics of what the configuration files are set up to do, which in practice really means, "in the logic of the tools" ... not in the use of XML per se.)

4. I cannot find any assertion in Manubens-Gil 2016 that cylc is more complicated when building workflows. If that statement really exists, then fine, but otherwise I'd remove this sentence.

5. I don't understand the first sentence of 2.2. and the statement "merely tools for task submission" ... (particularly in the case of cylc, when ROSE is used with it). (I am not here trying to argue in favour of cylc in any way, but simply to point out that one cannot describe these other things without being accurate about them!).

3. It appears that the magic sauce that makes this RCF/JUBE framework so useful is really in the layout and structure that is described in Figure 2 and within the various parameters defined in XML *and understood and actioned on by* JUBE and the RCF. The authors have moved material from the appendix to the main body, and that is helpful, but currently it reads like documentation, not an explanation of the functionality exposed. I think if the authors could find a better and more succinct way of explaining the functionality exposed by these configuration files the paper (and tools) would be vastly more interesting to prospective readers (and users).

1. E.g. page 22 states that "automatic archiving is performed", surely that is important functionality, and somehow configured ...

**Minor Issues**

1. The list of ways of generating compute demand (bottom of page 1) is still somewhat idiosyncratic: bundling data assimilation in with ensemble members doesn't make sense to me ... (especially since data assimilation appears to be an important option for ParaFlow - see end of page 14).

2. The paragraph on page 2 beginning line 11 doesn't seem really relate to the topic of the paper. If I were one of the authors I'd be arguing to remove it. The key points are in the next paragraph (but as I

said above, I can see an argument for removing much of this entire section).

3. The paranthetical comment on line 10 page 3 "(see article acknowledgement)" is not obviously pointing at the previous reference (I initially looked for something in the acknoweldgements of this manuscript, rather than the previously referenced paper). In any case, it's not just the size of the team, it's the time they spent as well.

4. I can't really see the segue between the last two paragraphs of section 1, probably because the authors have not really made clear to me what distinction exists between a workflow engine and a run control framework. I think I understand the that this tool is something which makes it a *specialised* workflow engine ... (the same issue exists with the last sentence on page 4).

5. The long paragraph which begins page 5 covers so many different things ... and introduces ParaFlow where generalised statements would be more appropriate.

6. Page 12/13 Eaton et al describes CF, not CMOR, and is inappropriately positioned at the end of the sentence (by appearing there it appears to be applying somehow to ParaFlow, not CF).

7. The second sentence of section 3.5 could be improved ... it's a very difficult sentence to deconstruct :-), and I think the use of the word exascale doesn't add anything.

---

## Author Response (AR2)

We thank the reviewer very much for going through this paper and making some great suggestions. We have hopefully addressed all of them and in doing so, have further improved the manuscript

**Reviewer 1:**
This paper presents a utility for controlling the execution and initial evaluation of an application (the ParFlow model) running in a (primarily) HPC framework. There are two levels to the framework described: the "run control framework" (or RCF) which itself utilises a more generic JUBE benchmark framework as a workflow engine. Essentially these provide a method of systematically defining, running and analysing some benchmarks - the authors also suggest it would be suitable for use for production simulations as well.

The paper is a heavily modified resubmission, and was previously titled: "Best practice regarding the three P's: profiling, portability and provenance when running HPC geoscientific applications".

The authors have made considerable efforts to respond to the previous reviews, and the paper is much improved, however there are still a number of issues. I think it could appear in GMD provided these issues are addressed to the satisfaction of the editor.

Most of my issues now reside in the front material, the body of the paper describing the use of the tool with paraview is much improved, and I am happy for that part to appear more or less as is.

**Major issues**

1. The goal of the paper is still not clear - but only because it is still obscured by what still feels like an excessive emphasis on motivation. I would ask the authors the following question: "If you read the abstract, what would you expect to find in the body of the paper?" Half the abstract is motivation, which seems wrong. Results of using this tool with ParaFlow are not even mentioned. Along the same lines, it is page 4 before the introduction gets round to telling us the bulk of the paper is about the RCF. These issues could relatively easily be addressed by reworking the abstract and either removing much of the existing introduction, or moving much of it to a motivation section immediately following.

We have removed some of the material from the introduction to shorten it and we've altered the abstract.

2. It is good to see there is now a discussion of other tools in section 2.1, but the material is not well connected, misses the point in a number of crucial ways, and (I would assert) wrong in some of the statements. There is no pre-existing taxonomy of tools in this space, and it would be unreasonable to expect the authors to have real experience with these tools, so getting the level of discussion right is not trivial, but

      1. Page 5, line 9 mashes a description of JUBE into a description of other tools. At the very least this is a new paragraph.

      We've now made this into a new paragraph.

      2. It is not clear why all the emphasis on XML. All XML provides is a syntax (which is obviously useful) but the statement that cylc has "its own scripting" (line 15) is mixing action (scripting) with the syntax definition (XML). Cylc actually uses INI and Jinja2 for syntax, what is interesting about the differences is not whether one uses XML or INI, but what semantics exist in the configuration. What can they do?

      This was trying to make the point that we've now already set up everything in xml and so we can easily switch to ecflow in that case so we've altered these lines to state this more explicitly.

3. The discussion about the platform constraints on submission belongs in its own paragraph (but I would ask why, with 2 hour limits, they can't run cylc - or any other tool - on a third system and simply poll through the login nodes using ssh tunnelling).

We would prefer to use something that runs natively on the machines we are using than having to set up a server to continuously run for the duration of our production job chains. Our production runs take 10 days of compute time (or more). We have tried to emphasize this point more.

3. No one expects such a comparison to assert that cylc or any other tool is better or worse than their tools, simply stating different capabilities is all that is really required. Clealy the JUBE RCF roadmap will differ from those tools, and it would be fine for them to describe and build other tools, even if they had the same or poorer functionalty - which is clearly not the case, there are some real advantages to describe here! The problem is that some of those advantages become clearer once the use case is fully described. It might be that this comparison could go in a paragraph preceding the conclusions, but whatever is done, all tools have strengths and weaknesses, this paragraph currently reads like a "defense" rather than a "comparison".

This also relates to the points below and above. We are trying to assert that we could use any of the tools mentioned- we chose JUBE as we had access to the developers and it is designed to run natively on the machines we use. The strengths in this case are that we do not have to run a separate server which continuously tunnels in over the duration of the run. We have tried to emphasize this but feel that listing all the strengths and weaknesses are beyond the scope of the paper as the focus is on the framework we built rather than the tools we use.

5. (Some of those advantages are actually in the semantics of what the configuration files are
set up to do, which in practice really means, "in the logic of the tools" ... not in the use of XML per se.)

Yes this is the point we were trying (unsuccessfully) to make. So we've tried to emphasize this- we've altered those lines and also altered the first sentence of 2.2.

4. I cannot find any assertion in Manubens-Gil 2016 that cylc is more complicated when building workflows. If that statement really exists, then fine, but otherwise I'd remove this sentence.

Quote: Cylc provides a cycling pattern that can be over just a sequence of integers or a powerful date-time cycle based on standard ISO 8601. It also uses the Jinja2 language to provide scripting features enhancing its workflow definition capabilities, although at the cost of a reduced readability. ecFlow's workflow definition language is not based on cycles, but it allows to define repetitions at job or family level. Moreover, it provides a Python API that makes the workflow definition easy, robust and powerful.

5. I don't understand the first sentence of 2.2. and the statement "merely tools for task submission" ... (particularly in the case of cylc, when ROSE is used with it). (I am not here trying to argue in favour of cylc in any way, but simply to point out that one cannot describe these other things without being accurate about them!).

We were trying to point out that it is really the RCF rather than the workflow engine that provides the useful infrastructure needed for complex geoscience applications. We have altered this sentence to try and reflect that.

3. It appears that the magic sauce that makes this RCF/JUBE framework so useful is really in the layout and structure that is described in Figure 2 and within the various parameters defined in XML and understood and actioned on by JUBE and the RCF. The authors have moved material from the appendix to the main body, and that is helpful, but currently it reads

like documentation, not an explanation of the functionality exposed. I think if the authors could find a better and more succinct way of explaining the functionality exposed by these configuration files the paper (and tools) would be vastly more interesting to prospective readers (and users).

      1.      E.g. page 22 states that "automatic archiving is performed", surely that is important functionality, and somehow configured ...

We're unsure what is meant by this point. We intended to describe the framework in the methods and then show the functionality in the case study.

**Minor Issues**

1. The list of ways of generating compute demand (bottom of page 1) is still somewhat idiosyncratic: bundling data assimilation in with ensemble members doesn't make sense to me ... (especially since data assimilation appears to be an important option for ParaFlow - see end of page 14).

Ensembles generate demand by needing to run several instances of the model at once. We have added this in.

2. The paragraph on page 2 beginning line 11 doesn't seem really relate to the topic of the paper. If I were one of the authors I'd be arguing to remove it. The key points are in the next paragraph (but as I Minor Issues said above, I can see an argument for removing much of this entire section).

We've removed a lot of this section based on the advice given above.

3. The paranthetical comment on line 10 page 3 "(see article acknowledgement)" is not obviously pointing at the previous reference (I initially looked for something in the acknoweldgements of this manuscript, rather than the previously referenced paper). In any case, it's not just the size of the team, it's the time they spent as well.

The acknowledgements referred to are in that citation. The people mentioned in the acknowledgements are part of the development team. However to avoid confusion we have removed that comment.

4. I can't really see the segue between the last two paragraphs of section 1, probably because the authors have not really made clear to me what distinction exists between a workflow engine and a run control framework. I think I understand the that this tool is something which makes it a specialised workflow engine ... (the same issue exists with the last sentence on page 4).

We tried to make the distinction clearer between a RCF and a workflow engine in the last paragraph of section 1.

The long paragraph which begins page 5 covers so many different things ... and introduces ParaFlow where generalised statements would be more appropriate.

We've shortened this paragraph and made it more general.

6. Page 12/13 Eaton et al describes CF, not CMOR, and is inappropriately positioned at the end of the sentence (by appearing there it appears to be applying somehow to ParaFlow, not CF).

We have included the additional reference for CMOR and moved the position of the citation.

7. The second sentence of section 3.5 could be improved ... it's a very difficult sentence to deconstruct :- ), and I think the use of the word exascale doesn't add anything.

We have tried to improve this sentence.

####################################################################

We thank the reviewer very much for going through this paper and making some great suggestions. We have hopefully addressed all of them and in doing so, have further improved the manuscript

**Reviewer 2:**

P2 11 – p3 3: the references in this section are all relatively old and a little out of date (there is nothing beyond 2014 in a rapidly developing area). It would be useful to include some more recent references. For example, GP-GPU accelerators are now well beyond "10s of cores" (p 2 25) and their performance is currently in the 10s of TFLOP/s range (rather than 1 TFLOP/s, p2 24 - e.g. see the NVIDIA Titan – and that's a year or two old now!).
Also, a major current trend towards developing exascale machines is the exploitation of FPGAs for acceleration of both applications and communication (e.g. the recently funded EU EuroEXA project, https://euroexa.eu/, among others).

We've shortened this paragraph and included more recent references as well as mentioning the FPGAs.

P4 9: please make it clear here that the ParFlow case study will focus on its execution on the Juqueen machine at JSC.

We've added this in.

P6 3: change to -> "The directory structure for the RCF run harness…"

We've made the change.

P9 12: it would be useful to more clearly link the section starting here on "health examination" to the follow up in Sec. 3.3. I suggest adding a subtitle (unnumbered) "Health Check Protocol" to make the link explicit.

 We've added the subtitle.

P9 24/25: for completeness, define the symbols used in eqn 1. (i.e. $T\_1$ and $T\_N$) – other symbols in equations are defined.

We've defined the symbols used.

P13 Fig. 3 I spotted that coll_message_size does not appear in Appendix B!

Thanks for picking this up, we've added this in.